# A ventrolateral medulla-midline thalamic circuit for hypoglycemic feeding

B. Sofia Beas[1,6], Xinglong Gu[2,6], Yan Leng[1], Omar Koita [1], Shakira Rodriguez-Gonzalez[1], Morgan Kindel [1], Bridget A. Matikainen-Ankney[3], Rylan S. Larsen [4], Alexxai V. Kravitz [3,5], Mark A. Hoon [2✉] & Mario A. Penzo [1✉]

Marked deficits in glucose availability, or glucoprivation, elicit organism-wide counter-regulatory responses whose purpose is to restore glucose homeostasis. However, while catecholamine neurons of the ventrolateral medulla (VLM$^{CA}$) are thought to orchestrate these responses, the circuit and cellular mechanisms underlying specific counter-regulatory responses are largely unknown. Here, we combined anatomical, imaging, optogenetic and behavioral approaches to interrogate the circuit mechanisms by which VLM$^{CA}$ neurons orchestrate glucoprivation-induced food seeking behavior. Using these approaches, we found that VLM$^{CA}$ neurons form functional connections with nucleus accumbens (NAc)-projecting neurons of the posterior portion of the paraventricular nucleus of the thalamus (pPVT). Importantly, optogenetic manipulations revealed that while activation of VLM$^{CA}$ projections to the pPVT was sufficient to elicit robust feeding behavior in well fed mice, inhibition of VLM$^{CA}$–pPVT communication significantly impaired glucoprivation-induced feeding while leaving other major counterregulatory responses intact. Collectively our findings identify the VLM$^{CA}$–pPVT–NAc pathway as a previously-neglected node selectively controlling glucoprivation-induced food seeking. Moreover, by identifying the ventrolateral medulla as a direct source of metabolic information to the midline thalamus, our results support a growing body of literature on the role of the PVT in homeostatic regulation.

[1] Unit on the Neurobiology of Affective Memory, National Institute of Mental Health, Bethesda, MD, USA. [2] Molecular Genetics Section, National Institute of Dental and Craniofacial Research, Bethesda, MD, USA. [3] National Institute of Diabetes and Digestive and Kidney Diseases, Bethesda, MD, USA. [4] Allen Institute for Brain Science, Seattle, WA, USA. [5] Department of Psychiatry, Washington University School of Medicine, St. Louis, MO, USA. [6] These authors contributed equally: B. Sofia Beas, Xinglong Gu. ✉email: mark.hoon@nih.gov; mario.penzo@nih.gov

Glucose is the main metabolic source of energy in the bloodstream and as such its level is tightly regulated. To ensure homeostatic control of this vital metabolite, large decreases in glucose availability – termed glucoprivation – elicit well-characterized counterregulatory responses (to reestablish normal blood glucose levels) that include mobilization of stored glucose, corticosterone secretion, and feeding behaviors[1,2]. Decades of research have shown that this system-wide response to glucoprivation is coordinated by centrally organized neural mechanisms. In particular, research indicates that catecholamine neurons of the ventrolateral medulla (VLM^CA) contribute to glucose counterregulatory responses[3]. Indeed, VLM^CA neurons are rapidly activated following episodes of extreme glucose deficiency. In turn, the VLM^CA is thought to trigger complex autonomic, endocrine and behavioral responses. Importantly, while both ascending and descending projections of VLM^CA neurons have been tied to specific counterregulatory responses[4,5], the precise downstream neural circuits controlling these responses, and in particular glucoprivation-induced feeding, remain unknown.

## Results

**VLM^CA–pPVT connections are sufficient to trigger food seeking.** To investigate putative circuits involved in glucoprivic feeding, we first examined the distribution of forebrain projections of VLM^CA neurons. For this, we labeled synaptic terminals arising from VLM^CA neurons by injecting an adeno-associated viral vector (AAV) expressing Synaptophysin-green fluorescent protein (Syp)-GFP in the VLM of TH-Cre mice, in which Cre recombinase expression is under the control of the tyrosine hydroxylase (TH) endogenous promoter[6,7] (Fig. 1a). This strategy allowed us to identify brain areas receiving synaptic inputs from VLM^CA neurons. Consistent with previous studies, we observed that VLM^CA neurons provide major synaptic input to multiple TH-rich forebrain structures, including the paraventricular nucleus of the hypothalamus, the bed nucleus of the stria terminalis, and the posterior portion of the paraventricular nucleus of the thalamus (pPVT)[8–10] (Fig. 1b, c). Of these projections we focused our studies on the pPVT due to its strong innervation of

the nucleus accumbens (NAc), a structure well-known to have major effects on reward behaviors including food seeking (Fig. 1a–c)[11,12].

Previous studies suggest that the PVT is a critical node linking hypothalamic hunger signals to goal-directed seeking behavior via projections to the ventral striatum[11,13]. However, whether the relay of metabolic status to the PVT is limited to hypothalamic input or may also occur through projections from hindbrain regions remains unclear. Consistent with previous literature, a recent study reported that chemogenetic activation of glucoprivation-sensitive VLM^CA neurons drives food intake in well-fed mice[6]. While chemogenetic activation is a useful tool to investigate the role of defined neurons, it has poor temporal resolution. Therefore, since VLM^CA neurons send projection to the pPVT, we sought to determine if these are required for food-seeking behavior by using optogenetics, which facilitates the dissection of different neural outputs with improved spatial resolution and temporal precision[14]. To accomplish this, we injected into the VLM of TH-Cre mice a Cre-dependent AAV-driving expression of Channelrhodopsin-2 (ChR2) (or as control, a fluorescent reporter) and stimulated these specific projections through an implanted optical fiber above the pPVT (Fig. 2a, b). Surprisingly, short optogenetic stimulation of ChR2-expressing VLM^CA–pPVT terminals was sufficient to drive robust feeding in well-fed mice (Fig. 2c–e and Supplementary Movie 1). As expected, this behavioral phenotype was strictly associated with ChR2 expression because light-evoked feeding was absent in control subjects (Fig. 2f, g). These data suggest that a pPVT downstream circuit may be part of a specific VLM-mediated response pathway.

While our finding that the pPVT is involved in food seeking is consistent with recent publications[11,15], activity in the pPVT has also been linked to the emergence of negative emotional states such as anxiety and aversion[16]. Thus, to investigate whether VLM^CA–pPVT projections might also impact these processes, we assayed mice where we stimulated VLM-projections in the elevated zero maze (EZM), the open field test (OFT), and the real-time place preference (RTPP) assay (Supplementary Fig. 1).

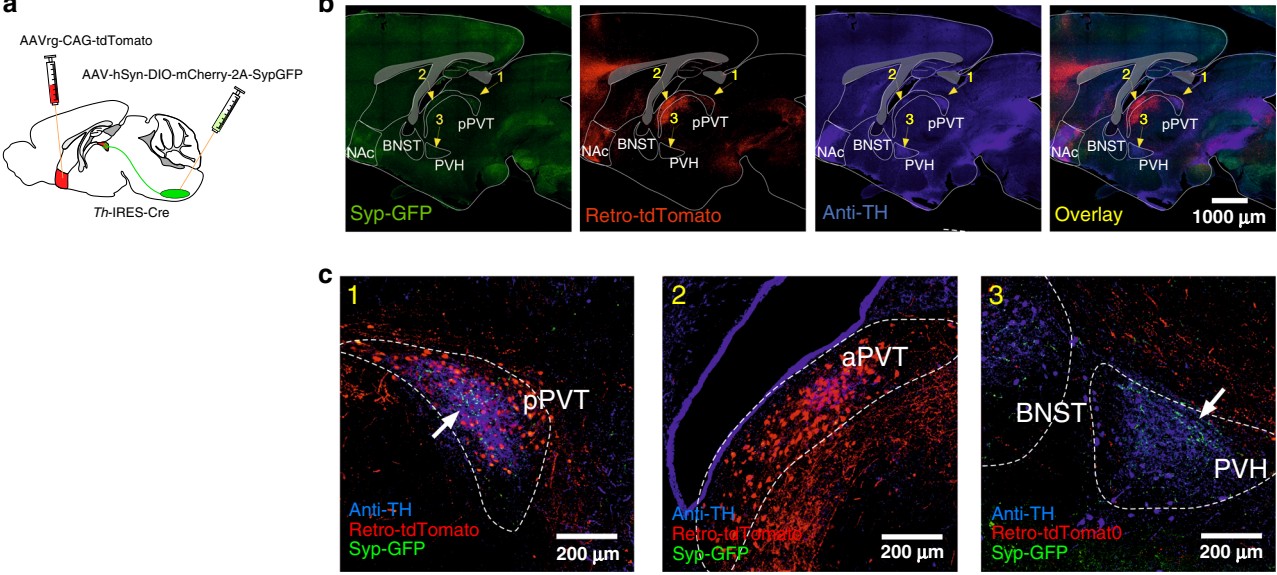

**Fig. 1 Axonal projections from VLM^CA neurons innervate the pPVT. a** Schematic of virus injections strategy used to examine putative forebrain areas that had robust axonal inputs of VLM^CA neurons and that project to the NAc. **b** Representative images of a sagittal brain slice showing the distribution of VLM^CA projections, NAc-projecting neurons (Retro-tdTomato) and TH immunoreactivity (anti-TH) across various forebrain regions. **c** Representative images showing magnifications of the areas depicted in **b**, namely the pPVT (1), the aPVT (2), and the BNST/PVH region (3). White arrows depict regions of dense Syp-GFP expression.

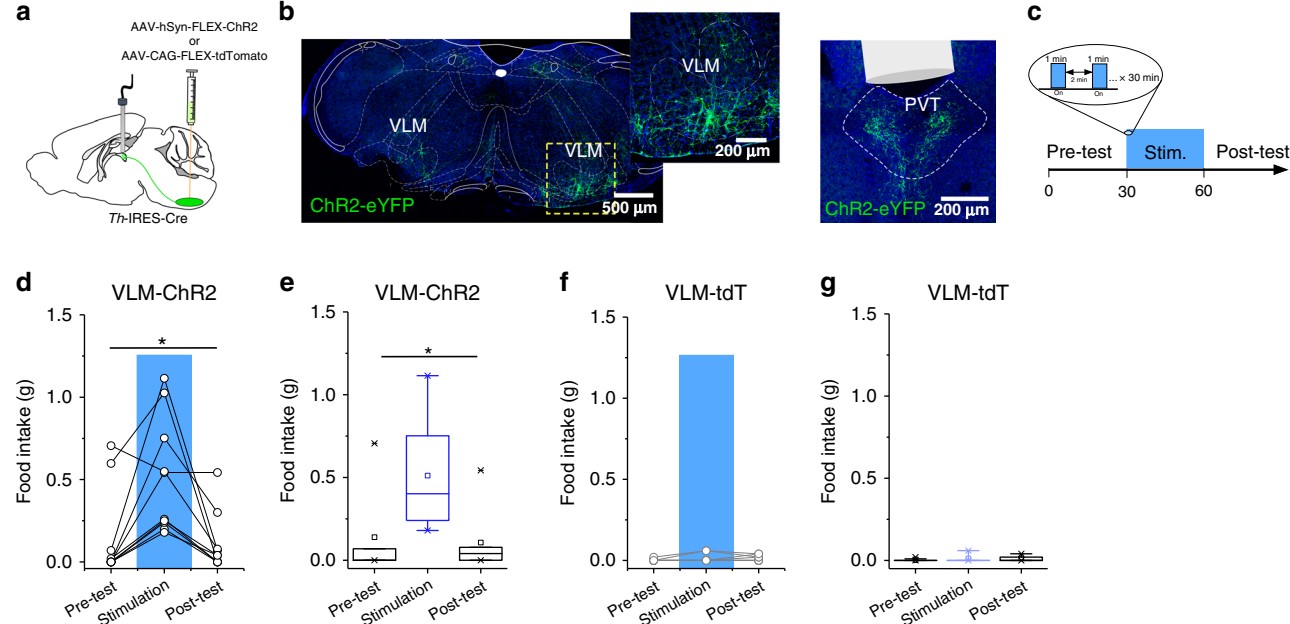

**Fig. 2 Optogenetic stimulation of the VLM$^{CA}$ terminals in the pPVT results in robust feeding behavior in well-fed mice. a** Schematic of ChR2-expressing and control viral vector injections, as well as optical fiber implantation for optogenetic activation VLM$^{CA}$ fibers in the pPVT. **b** Representative image of ChR2 expression in the VLM of TH-IRES-Cre mice and ChR2-labeled terminals in the pPVT. **c** Schematic of the experimental design and the stimulation protocol used to measure feeding behavior during optogenetic stimulation of VLM terminals in the pPVT. Food intake was quantified in well-fed mice prior to, during, and after light stimulation (30 min, pre-test, stimulation, and post-test). The stimulation protocol consisted of 30 min in which light stimulation alternated between 1 min "light ON" (20 Hz) and 2 min "light OFF" bouts. **d** Plot of individual performance on feeding behavior during the pre-test, stimulation, and post-test for ChR2 (VLM-ChR2) mice. **e** Quantification of feeding behavior during VLM$^{CA}$–pPVT stimulation for ChR2 mice. Total food intake in grams, ChR2: Pre-Test, 0.116 ± 0.07; Stimulation, 0.430 ± 0.11; Post-Test, 0.115 ± 0.05, $n = 12$ mice; $F_{(2,33)} = 5.12$, one-way ANOVA followed by Tukey's test. Group comparisons: Pre-Test vs Stimulation, *$P = 0.02$; Post-Test vs Stimulation, *$P = 0.02$. **f** Plot of individual performance on feeding behavior during the pre-test, stimulation, and post-test for control (VLM-tdT) mice. **g** Quantification of feeding behavior during VLM$^{CA}$–pPVT stimulation for control mice. Total food intake in grams, Control: Pre-Test, 0.002 ± 0.01; Stimulation, 0.012 ± 0.03; Post-Test, 0.012 ± 0.02, $n = 10$ mice; $F_{(2,27)} = 1.04$, one-way ANOVA followed by Tukey's test. Group comparisons: Pre-Test vs Stimulation, $P = 0.44$; Post-Test vs Stimulation, $P = 1$. Box chart legend: box is defined by 25th, 75th percentiles, whiskers are determined by 5th and 95th percentiles, and mean is depicted by the square symbol. Data presented as mean ± SEM.

Results of these experiments showed that the VLM$^{CA}$–pPVT pathway was associated with decreased exploration of the open segments of the EZM, consistent with an anxiogenic role of this projection (Supplementary Fig. 1a–c). Anxiogenic effects were not as apparent in the OFT, where instead a significant reduction in total distance traveled was observed (Supplementary Fig. 1d–g). Finally, optogenetic stimulation did not alter time spent in the stimulated side in RTPP assays (Supplementary Fig. 1h–k). However, a similar reduction in locomotor activity, as seen in the OFT, was observed in this task (Supplementary Fig. 1k). These results indicate that while stimulation of VLM$^{CA}$–pPVT projections does not causes aversion, activity in this pathway is associated with increased anxiety and reduced exploratory behavior[17]. Altogether, our findings suggest that VLM$^{CA}$ projections to the pPVT drive robust feeding behavior in mice, with moderate effects on affect.

**pPVT–NAc projecting neurons are activated by VLM$^{CA}$ input.** Anatomical studies have found that most neurons of the pPVT innervate the NAc[18]. In addition, recent studies implicate PVT–NAc connectivity in reward-related behaviors in rodents[11,12,19]. Based on these findings, we hypothesized that VLM$^{CA}$ neurons promote feeding behavior by modulating the activity of pPVT neurons projecting to the NAc. To test this prediction, we first examined whether VLM$^{CA}$ neurons form anatomical connections with NAc-projecting neurons of the pPVT using monosynaptic tracing[20]. With this approach, we

found that TH-expressing VLM neurons monosynaptically innervate NAc-projecting PVT neurons (Supplementary Fig. 2). Next, to probe whether VLM$^{CA}$ projections form functional connections onto PVT–NAc neurons, we optogenetically stimulated VLM$^{CA}$–PVT projections (using a red-shifted ChR2 variant; ChrimsonR) and imaged calcium responses in NAc-projecting pPVT neurons using fiber photometry in awake animals (Fig. 3a–c). To achieve selective expression of the genetically encoded calcium sensor GCaMP6s in NAc-projecting neurons of the pPVT, we injected a viral vector that drives retrograde expression of Cre recombinase (AAV(retro)-CAG-Cre) bilaterally in the NAc of TH-Cre mice and subsequently a vector encoding Cre-dependent GCaMP6s in the pPVT. Interestingly, we observed that optogenetic activation of VLM$^{CA}$ input significantly increased the fluorescence of the genetically encoded calcium sensor GCaMP6s in NAc-projecting pPVT neurons (Fig. 3d–g). Notably, this effect was dependent on VLM$^{CA}$ neuron expression of ChrimsonR (Fig. 3f, g).

It is important to highlight that a limitation of this approach is the possibility that delivering AAV(retro)-CAG-Cre in the NAc might result in unwanted expression of Cre recombinase in non-TH neurons of the VLM of TH-Cre mice. Thus, while to the best of our knowledge a functional VLM input to the NAc has not been described in the literature, we conducted a variation of our initial approach wherein expression of GCaMP6s depended instead on Flp-mediated recombination (Supplementary Fig. 3). Using this alternative strategy, we confirmed that stimulation of

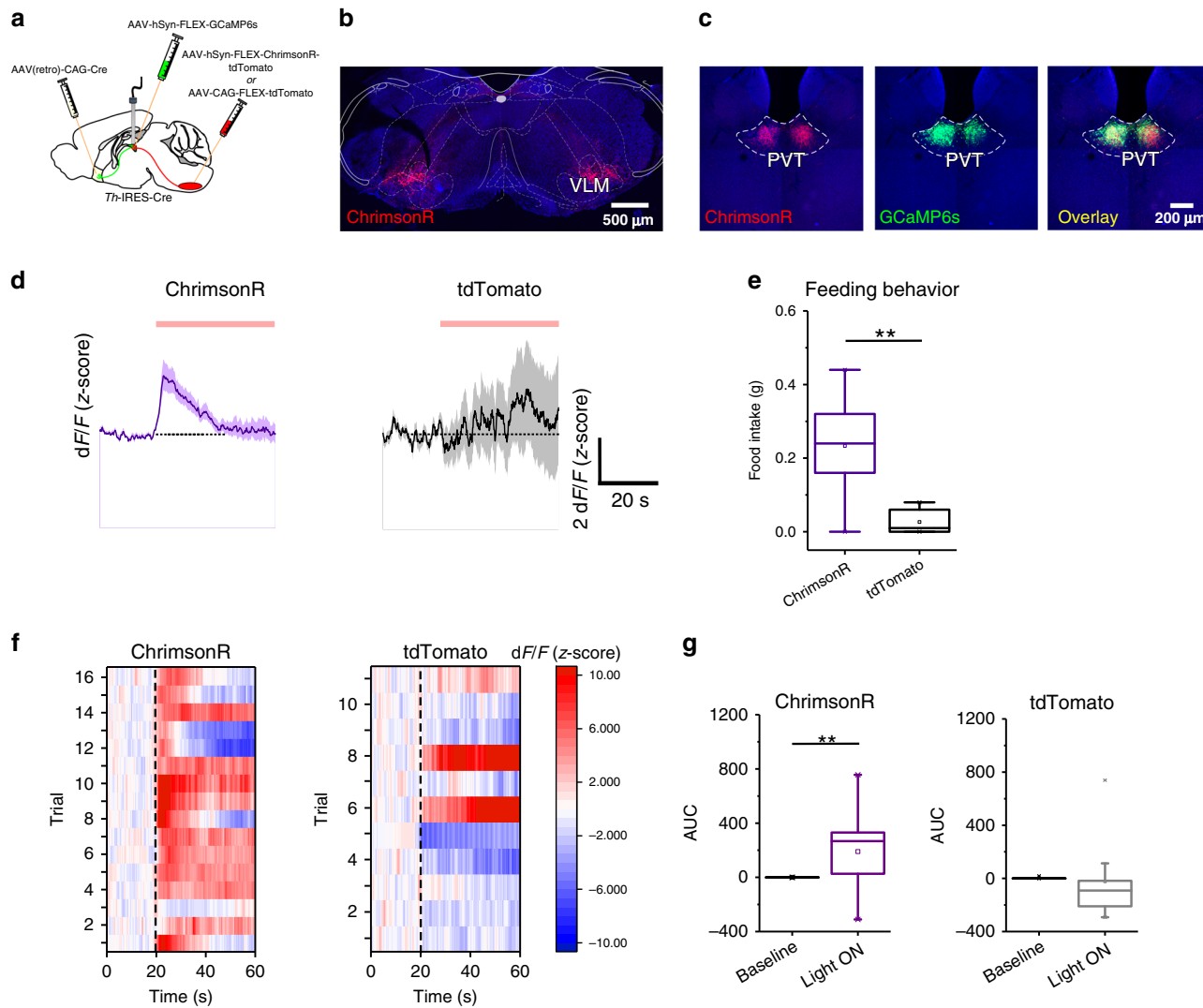

**Fig. 3 Photostimulation of VLM^CA axonal inputs to pPVT increases the activity of pPVT–NAc neurons. a** Schematic of the viral vector strategy and optical fiber placement used for selectively expressing GCaMP6s in NAc-projecting neurons of the pPVT and red-shifted channelrhodopsin-2 (ChrimsonR-tdTomato) in catecholaminergic neurons of the VLM of TH-IRES-Cre mice. **b** Representative images from a TH-IRES-Cre mouse expressing ChrimsonR-tdTomato in the VLM. **c** Representative images of the pPVT showing axonal projections from the VLM (ChrimsonR-tdTomato; left), GCaMP6s expression in pPVT–NAc neurons (middle), and overlay image showing colocalization between the projections from ChrimsonR-expressing VLM neurons and GCaMP6s-expressing pPVT–NAc neurons (right). **d** Average GCaMP6s responses from pPVT–NAc neurons in ChrimsonR-expressing (purple) and tdTomato-expressing (black) animals subjected to light stimulation. **e** Quantification of food intake during VLM^CA–pPVT stimulation for both ChrimsonR and tdTomato showing increases in feeding after stimulation in well-fed ChrimsonR expressing mice. Total food intake in grams, ChrimsonR, 0.23 ± 0.05, $n = 9$ mice; tdTomato, 0.03 ± 0.01, $n = 6$ mice; two-sided Unpaired t-test, **$P = 0.006$. **f** Heatmap showing individual trial GCaMP6s responses to light stimulation (561 nm at 20 Hz) from both ChrimsonR-expressing and tdTomato-expressing (control) mice. **g** Left: Quantification of light-evoked changes in GCaMP6s fluorescence in pPVT–NAc neurons from mice expressing ChrimsonR in VLM^CA neurons. Area under the curve (AUC), Baseline, −0.21 ± 0.11; Light ON, 190.65 ± 63.27; two-sided Paired t-test, **$P = 0.008$. Right: Quantification of light-evoked changes in GCaMP6s fluorescence in pPVT–NAc neurons from mice expressing tdTomato in VLM^CA neurons. Area under the curve (AUC), Baseline, 1.78 ± 1.83; Light ON, −23.13 ± 99.86; two-sided Paired t-test, $P = 0.79$. Box chart legend: box is defined by 25th, 75th percentiles, whiskers are determined by 5th and 95th percentiles, and mean is depicted by the square symbol. Data presented as mean ± SEM.

VLM^CA inputs to the pPVT leads to significant increases in GCaMP6s fluorescence (Supplementary Fig. 3a–d) and drives robust feeding behavior (Supplementary Fig. 3e). Collectively, these findings demonstrate that activation of the VLM^CA–pPVT pathway is associated with increases in pPVT neuronal activity and a robust feeding phenotype.

We employed two further strategies to obtain evidence supporting this VLM^CA–pPVT–NAc circuit. First, using chemogenetics, we found that stimulation of VLM^CA neurons markedly increased the expression of the immediate early gene cFos in pPVT neurons (Supplementary Fig. 4). Second, we performed in vitro whole-cell patch-clamp recordings from unidentified neurons of the pPVT – since most pPVT neurons project to the NAc[18] – and stimulated ChR2-expressing terminals from VLM^CA neurons in this region. We observed that optogenetic stimulation of VLM^CA afferents leads to small but reproducible depolarizations in a large fraction of pPVT neurons (Supplementary Fig. 5). Importantly, this effect was prevented by bath application of the beta-adrenergic receptor blocker propranolol, consistent with the adrenergic nature of VLM^CA projections[21,22]. In summary, these

results lend further support to the idea that VLM[CA] projections excite pPVT–NAc neurons.

Our observation that VLM[CA]–pPVT projections, which promote feeding in well-fed mice, excite pPVT–NAc neurons is at odds with some studies linking PVT function to the consumption of natural rewards. Indeed, previous reports have linked feeding behavior with inhibitory modulation of the PVT and the PVT–NAc pathway[23–26]. However, others have shown that hunger signals and food seeking are associated with increases in the activity of PVT neurons[19,27–29]. One interpretation that may reconcile these seemingly contradictory results is that PVT neurons play dissociable roles in food seeking and consummatory behaviors through their NAc projections, such that increases in PVT–NAc activity promote food-seeking behavior, while decreases in neuronal activity result in consummatory behavior. This idea is partially supported by a recent report showing that optogenetic stimulation of a subpopulation of NAc-projecting PVT neurons promotes food-seeking behavior[19], and by the observation that reward consumption is associated with inhibition of PVT–NAc projections[11,26]. Collectively, these studies support the idea that bidirectional modulation of PVT circuits coordinates reward seeking and consummatory (feeding) behaviors. However, because this model has not been formally tested, we sought to investigate the dynamics of pPVT–NAc neuronal activity during both the approach and consummatory phases of a reward foraging decision task (Fig. 4). Specifically, mice were trained to navigate a linear track in which visiting a "trigger-zone" located on one end of the track enabled access to a reward on the opposite end ("reward-zone") (Fig. 4b). For each trial, reward availability was signaled via a 1 s 8 KHz tone presented when subjects remained on the "trigger-zone" for a least 2 s. Mice readily learned to perform this task, as exemplified by their ability to complete 40–50 trials per session by the end of training. Next, using fiber photometry we assessed the responses of pPVT–NAc neurons during both reward approach (as a measurement of food-seeking behavior) and consumption (see Methods). Consistent with the above model, we observed that while reward approach was associated with a prominent increase in GCaMP6s fluorescence in pPVT–NAc neurons, reward access and consumption were largely associated with an abrupt decrease in fluorescence (Fig. 4c–h). Collectively, these results demonstrate that the pPVT–NAc pathway is bidirectionally modulated by reward seeking and consumption.

To investigate whether similar dynamics of pPVT–NAc neuronal activity follows VLM[CA]-mediated activation of this circuit, we contrasted the effects of light stimulation with those of subsequent feeding events. Specifically, we analyzed changes in the GCaMP6s fluorescence of pPVT–NAc neurons following the retrieval of individual food pellets (see Methods) from an automated food dispenser[30]. This analysis revealed that while activation of VLM[CA] projections led to short latency increases in neuronal activity in the pPVT, feeding itself was associated to marked reductions in GCaMP6s fluorescence (Fig. 5 and Supplementary Figs. 3f–h and 6). Importantly, these food-associated attenuations in calcium signals were distinct and temporally dissociated from those induced by the cessation of light stimulation (Supplementary Fig. 7). In summary, our findings are consistent with a model wherein activation of pPVT–NAc neurons by VLM[CA] projections drives food seeking, but subsequent feeding attenuates activity in this pathway. Interestingly, light-induced excitation of pPVT–NAc neurons was significantly greater in sessions in which access to food was prevented during the test (Supplementary Fig. 7). Considering the anxiogenic and aversive effects associated with activation of the pPVT, these findings suggest that activation of VLM[CA] projections to the pPVT may drive food seeking through negative reinforcement.

**The VLM[CA]–pPVT–NAc circuit is required for glucoprivation-induced food seeking.** Given the known role of VLM[CA] neurons in glucoprivation-mediated feeding behavior, we sought to determine whether VLM[CA]–pPVT communication was involved in this process. For this, we used the glucose analogue 2-deoxy-D-glucose (2DG) – which inhibits glycolysis – to produce a model of glucoprivic feeding (Supplementary Fig. 8). Consistent with previous studies[2], intraperitoneal (i.p.) injections of 2DG led to a robust increase in cFos immunoreactivity in VLM[CA] neurons (Supplementary Fig. 9). Notably, this manipulation led to similar increases in cFos expression in the pPVT (Supplementary Fig. 9). Next, to investigate whether the 2DG-mediated increase in cFos expression in the pPVT depended on VLM[CA] neuronal activity, we chemogenetically silenced VLM[CA] neurons with an inhibitory *designer receptor exclusively activated by designer drug* (DREADD), during glucoprivation (Fig. 6a, b). As predicted, i.p. injection of the DREADD ligand clozapine-*N*-oxide (CNO) prior to 2DG administration significantly attenuated cFos expression in the pPVT when compared to saline-injected controls (Fig. 6c, d). These effects were not observed in mice in which VLM[CA] neurons were infected with control AAV (Fig. 6c, d). Together, these results demonstrate that glucoprivation-associated activation of pPVT neurons requires VLM[CA] input. A corollary of this finding is that glucoprivation (2DG)-induced feeding should require activation of the VLM[CA]–pPVT pathway. To verify this, we optogenetically silenced VLM[CA] inputs to the pPVT immediately following 2DG administration using the light gated chloride ion-pump, eNpHR (halorhodopsin) (Fig. 7a, b). Congruent with our proposition, silencing VLM[CA]–pPVT projections significantly attenuated 2DG-induced increases in feeding behavior (Fig. 7c). Notably, optogenetic silencing of VLM[CA] axons to the pPVT had no effect on 2DG-mediated adrenal medullary hyperglycemia or increases in corticosterone secretion (Fig. 7d, e). These findings lend empirical support to the idea that functional segregation exists across the various projections of VLM[CA] neurons[2]. In addition, we did not observe any effect of optogenetically silencing VLM[CA]–pPVT projections on feeding behavior elicited by acute food restriction in the absence of 2DG (Fig. 7f), further highlighting that this circuit is specifically engaged during glucoprivic events and is not involved in regulating normal food intake associated with acute food restriction[31].

Lastly, we examined whether the pPVT–NAc pathway is necessary for glucoprivic feeding. For this, we chemogenetically silenced pPVT–NAc projecting neurons using the inhibitory DREADD (Fig. 8a, b). Our results showed that chemogenetic inhibition of pPVT–NAc projecting neurons prior to 2DG administration markedly attenuated 2DG-induced feeding in DREADD-expressing mice (Fig. 8c). These findings underscore the importance of pPVT–NAc pathway activation during food-seeking behavior.

## Discussion
Here we have defined a previously neglected VLM[CA]–pPVT–NAc circuit that supports and mimics glucoprivation-induced feeding behavior (Supplementary Fig. 10). Our identification of this circuit provides a novel target for the pathogenesis of hypoglycemia-associated autonomic failure (HAAF), a potentially lethal condition marked by an inability to detect ensuing glucoprivic episodes. Notably, our findings are in stark contrast with a recent report suggesting that hypothalamic projections of catecholaminergic neurons of the nucleus of the solitary tract, which neighbors the VLM, guide glucoprivic feeding[32]. While the nature of this discrepancy is unclear, it is important to mention that whereas most studies on glucose counterregulatory mechanisms attribute glucoprivation-induced feeding to the function of

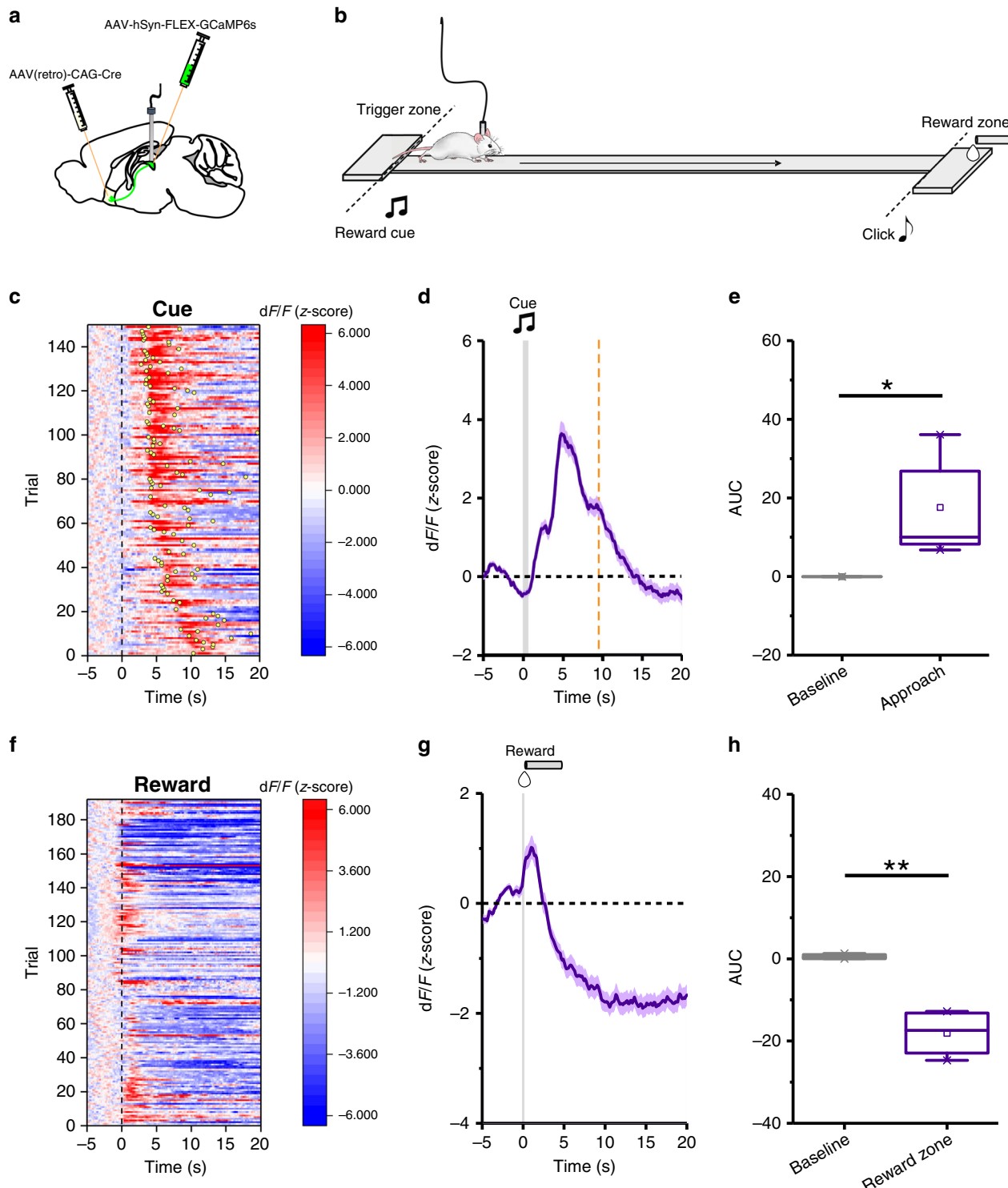

VLM$^{CA}$ neurons[2], the NTS has been primarily reported to promote anorexinergic responses[33–36].

While these findings coincide with previous studies highlighting a role for the PVT in both aversive and reward-related behaviors[11,16,19,37–44], the complex nature of the behavioral phenotype elicited by the activation of VLM$^{CA}$–pPVT projections warrant further discussion. Notably, whereas optogenetic stimulation of this pathway acutely decreases exploratory behavior and increases anxiety, feeding behavior emerges shortly after the offset of light stimulation (Supplementary Fig. 6; Supplementary Movie 1). One potential interpretation to these seemingly

contradicting behavioral outcomes is that activation of VLM$^{CA}$–pPVT projections mimics the debilitating and aversive effects of hypoglycemic state[45–47]. Indeed, previous studies have shown that VLM$^{CA}$ neurons, which encompass the noradrenergic (A1) and adrenergic (C1) cell groups[2], are activated by potentially life-threatening situations such as hypoxia and hypoglycemia[3,48] suggesting that their recruitment promotes aversive states[49]. In addition, administration of 2DG to induce hypoglycemia has been shown to[49,50] drive decreased locomotion and to promote negative affective states, thereby resembling the effects of VLM$^{CA}$–pPVT stimulation. Within this context, VLM$^{CA}$ neurons

**Fig. 4 The activity of pPVT–NAc neurons increases during food-seeking behavior in hungry mice and decreases during feeding. a** Schematic of the viral vector strategy and optical fiber placement used for selectively expressing GCaMP6s in NAc-projecting neurons of the pPVT. **b** Schematics of the reward foraging task. To initiate a trial, mice awaited in the trigger zone for an auditory cue (8 KHz, 1 s) which signaled reward availability. Mice then ran to the reward zone to retrieve a liquid food reward. **c** Heatmap of GCaMP6s responses from pPVT–NAc neurons during performance of individual trials of the foraging task. GCaMP6s responses were time-locked to cue onset and trials were sorted by latency to complete a trial. Yellow dots represent first nose-poke into the food-port. **d** Average GCaMP6s responses from pPVT–NAc neurons during the foraging task. Gray bar indicates the delivery of auditory cue during trial initiation. Trial inclusion criteria was set to trials performed under 10 s noted by the red dotted line. **e** Quantification of the changes in GCaMP6s fluorescence in pPVT–NAc neurons during the onset of trial performance in the foraging task. AUC, Baseline, $-0.05 \pm 0.02$; Approach, $17.58 \pm 5.87$; $n = 5$ mice, two-sided Paired t-test, $*P = 0.04$. **f** Heatmap of GCaMP6s responses from pPVT–NAc neurons during feeding bouts in the foraging task. GCaMP6s responses were time-locked to first nose-poke into the food-port. **g** Average GCaMP6s responses from pPVT–NAc neurons during feeding bouts in the foraging task. Gray line signals first nose-poke into the food-port. **h** Quantification of the changes in GCaMP6s fluorescence in pPVT–NAc neurons during the onset of feeding in the foraging task. AUC, Baseline, $0.52 \pm 0.31$; Approach, $-18.08 \pm 2.91$; $n = 4$ mice, two-sided Paired t-test, $**P = 0.008$. Box chart legend: box is defined by 25th, 75th percentiles, whiskers are determined by 5th and 95th percentiles, and mean is depicted by the square symbol. Data presented as mean ± SEM.

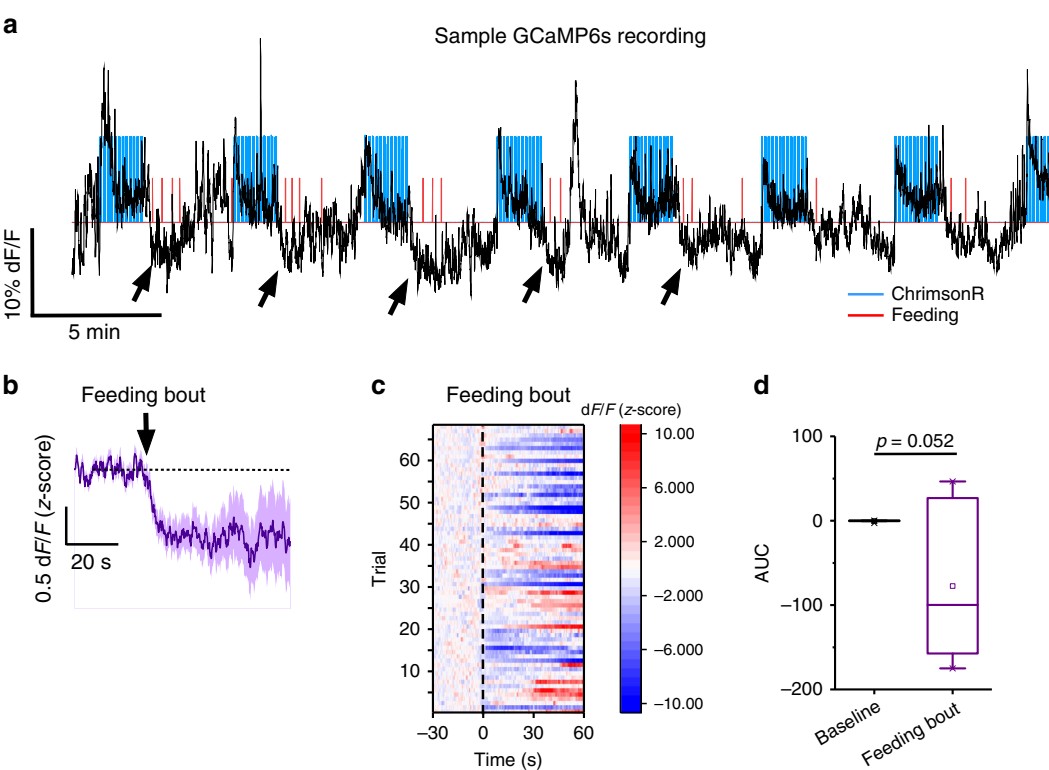

**Fig. 5 Feeding behavior is associated with decreases in GCaMP6s fluorescence in pPVT–NAc neurons. a** Representative trace showing the effect of light stimulation (blue lines) on GCaMP6s fluorescence of pPVT–NAc neurons (black line). Notice that initiation of food intake (red lines) was immediately followed by decreases in the fluorescence of the GCaMP6s signal (depicted with black arrowheads). **b** Average GCaMP6s response from pPVT–NAc neurons following the onset of a feeding bout (illustrated by the black arrow). **c** Heatmap of individual trials GCaMP6s responses to the initiation of feeding bouts from ChrimsonR-expressing mice. **d** Quantification of feeding bouts onset changes in GCaMP6s fluorescence in pPVT–NAc neurons. AUC, Baseline, $-0.47 \pm 0.38$; Feeding bout, $-77.48 \pm 35.16$; $n = 7$ mice, two-sided Paired t-test, $P = 0.052$. Box chart legend: box is defined by 25th, 75th percentiles, whiskers are determined by 5th and 95th percentiles, and mean is depicted by the square symbol. Data presented as mean ± SEM.

may relay hypoglycemia-related aversive signals to the pPVT to promote food-seeking behavior. This notion is consistent with extensive literature demonstrating that the PVT is sensitive to aversive stimuli and participates in regulating adaptive behaviors[16,51]. Moreover, our results resemble recent observations linking the aversive elements of drug withdrawal to increased PVT–NAc activity and subsequent drug-seeking behavior[40]. Indeed, here we have included novel evidence demonstrating that while food-seeking behavior is associated with the activation of PVT–NAc projections, subsequent reward consumption results in inhibition of this pathway (Fig. 4). The latter observation is consistent with a recent report showing that non-appetitive rewarding stimuli attenuate PVT neuronal activity[52]. Collectively, these findings raise the possibility that the PVT drives goal-directed behavior through negative reinforcement. Indeed, while not directly assessed in the present study, our observation that the activational effects of VLM$^{CA}$–pPVT stimulation are potentiated when animals do not have access to food is consistent with this model (Supplementary Fig. 7).

It is important to consider that since fiber-based photometric signals represent the collective activity of a group of cells (in this case NAc projecting pPVT neurons), it is possible that pPVT–NAc neurons driving food seeking could be distinct from those modulated by food consumption. Similarly, PVT neurons promoting reward-related behaviors may be distinct from those that participate in the orchestration of defensive behaviors[37,38].

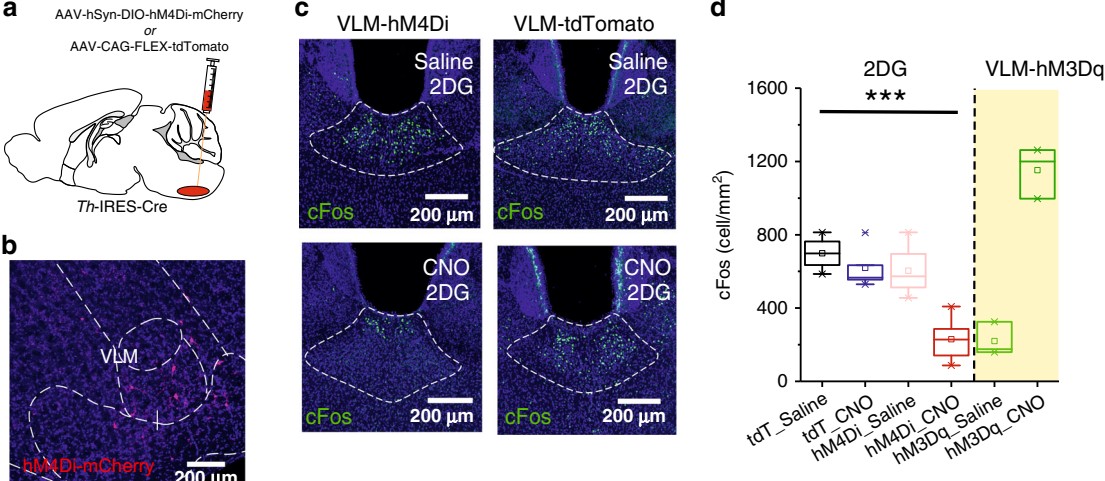

**Fig. 6 VLM$^{CA}$ neurons mediate glucoprivation-induced activation of the pPVT. a** Schematic of the stereotaxic injections for selective expression of DREADDi (hM4Di-mCherry) in VLM$^{CA}$ neurons of TH-IRES-Cre mice. **b** Representative images of the VLM from a TH-IRES-Cre mouse expressing the hM4Di-mCherry virus. **c** Representative images showing cFos immunohistochemistry in the pPVT of mice expressing either the hM4Di-mCherry (left) or tdTomato control virus (right) in the VLM$^{CA}$ and injected with either saline (top) or with CNO (bottom), 30 min before administration of 2DG. **d** Quantification of cFos immunoreactivity in PVT as cells/mm², tdT-Saline, 698.85 ± 46.88, $n = 6$ mice; tdT-CNO, 619.06 ± 51.00, $n = 6$ mice; hM4Di-Saline, 603.80 ± 53.13, $n = 6$ mice; hM4Di-CNO, 229.65 ± 46.19, $n = 6$ mice; effect of treatments, $F_{(1,17)} = 19.70$, $P < 0.001$; effect of virus, $F_{(1,17)} = 22.43$, $P < 0.001$; interaction, $F_{(1,17)} = 8.28$, $P < 0.05$, two-way analysis of variance (ANOVA) followed by Tukey's test. Group comparisons: tdT_Saline vs hM4Di_CNO, ***$P = 0.00004$, tdT_CNO vs hM4Di_CNO, ***$P = 0.0002$, hM4Di_Saline vs hM4Di_CNO, ***$P = 0.0002$. Data from Supplementary Fig. 4 included for comparison purposes (shaded portion of the graph). Box chart legend: box is defined by 25th, 75th percentiles, whiskers are determined by 5th and 95th percentiles, and mean is depicted by the square symbol. Data presented as mean ± SEM.

Thus, as recently proposed, future studies should aim to determine how functionally dissociable PVT neuronal subclasses are[53]. Indeed, functionally distinct cell types have been recently described in the PVT[52].

A growing body of literature supports the idea that the PVT is an important regulator of reward-related behaviors[11,12,23,27,39,40,54]. Interestingly, PVT's role in reward seeking appears to be under the control of a wide range of afferent pathways[55]. While the precise contributions of these PVT afferent inputs to the regulation of appetitive behaviors have not been fully elucidated, current evidence suggest distinct roles of these inputs. The presence of such distinct inputs coincides with an emerging view of the PVT as a major hub for interoceptive and exteroceptive information integration[16,56]. As such, whereas input from the prefrontal cortex appears to relay information about learned associations to the PVT (e.g., reward predictive cues)[11,54], other projections and circuit specializations more specifically convey information about energetic needs[19,25,57]. An additional level of complexity is likely overlaid upon these distinct afferent pathways, namely state-dependent modulation of neuronal activity in the PVT. For example, orexinergic input to the PVT may carry arousal-related signals that invigorate reward-seeking behavior[12,58]. In contrast, inputs relaying information about threatening situations may dampen PVT responding to reward-related cues resulting in diminished food seeking[43,59,60]. This view is supported by recent studies suggesting that the PVT is required amid motivational conflicts for the integration of reward- and aversive-related guiding behavior[42,43].

While conceptual models of the function of the VLM are far from complete, decades of research have identified the existence of distinct VLM$^{CA}$ neuronal subgroups that are differentially activated by glucoprivation. Thus, while mild glucoprivation is known to drive hyperglycemia via the activation of spinally projecting VLM$^{CA}$ neurons, feeding behavior is selectively induced under intense glucoprivation[61] and requires the activity of forebrain-projecting VLM$^{CA}$ neurons[3]. A major forebrain target of these VLM$^{CA}$ neurons, the PVT, is thought to participate in driving food-seeking behavior. But most models assume that the PVT integrates metabolic signals arising exclusively from the hypothalamus[13]. In addition, a recent study demonstrated that PVT neurons are equipped with intrinsic mechanisms that may facilitate food seeking in response to decreases in extra-cellular glucose[19]. Whereas the biologically relevant circumstances that lead to direct recruitment of the PVT by this mechanism are not fully understood, our study expands the role of the PVT in metabolic regulation, by showing that, under extreme circumstances, metabolic information can be directly communicated to it by hindbrain catecholaminergic cells. As such, in addition to the intrinsic glucose-sensing mechanisms of PVT neurons, the existence of parallel hypothalamus–PVT and VLM$^{CA}$–pPVT circuits may constitute fail-safe mechanisms that ensure appropriate adaptation to metabolic status. Future studies should aim define the precise circuit and cellular mechanisms by which the pPVT–NAc pathway coordinates reward-related behaviors.

## Methods

**Mice.** All procedures were approved by the National Institute of Mental Health Animal Care and Use Committees performed following the Guide for the Care and Use of Laboratory Animals. For all experiments, both male and female (7–12 weeks of age), *TH*-IRES-Cre mice (European Mouse Mutant Archive; stock number: EM:00254; backcrossed five generations with C57Bl/6NJ mice) and C57BL/6NJ strain mice (The Jackson Laboratory) were used in this study. Mice were housed under a 12-h light–dark cycle (6 a.m. to 6 p.m. light) at temperature of 70–74 °F and 40–65% humidity, with food and water available ad libitum. Animals were randomly allocated to the different experimental conditions reported in this study.

**Viral vectors.** AAV2/5-Ef1a-DIO hChR2(E123T/T159C)-EYFP and AAV9-hSyn-FLEX-GCaMP6s-WPRE-SV40 were produced by the Vector Core of the University of Pennsylvania. AAV9-EF1a-DIO-eNpHR3.0-mCherry, AA9-CAG-FLEX-tdTomato, AV9-hSyn-DIO-mCherry-2A-SyneGFP and AAV2-Syn-DIO-GFP were produced by the Vector Core of the University of North Carolina. AAV5-Syn-FLEX-ChrimsonR-tdTomato (Addgene plasmid # 62723) and AAV2-hSyn-DIO-hM4Di-mCherry, AAV2-hSyn-DIO-hM3D(Gq)-mCherry were produced by Addgene. AAV2(retro)-CAG-iCre (Addgene plasmid # 81070) was produced by

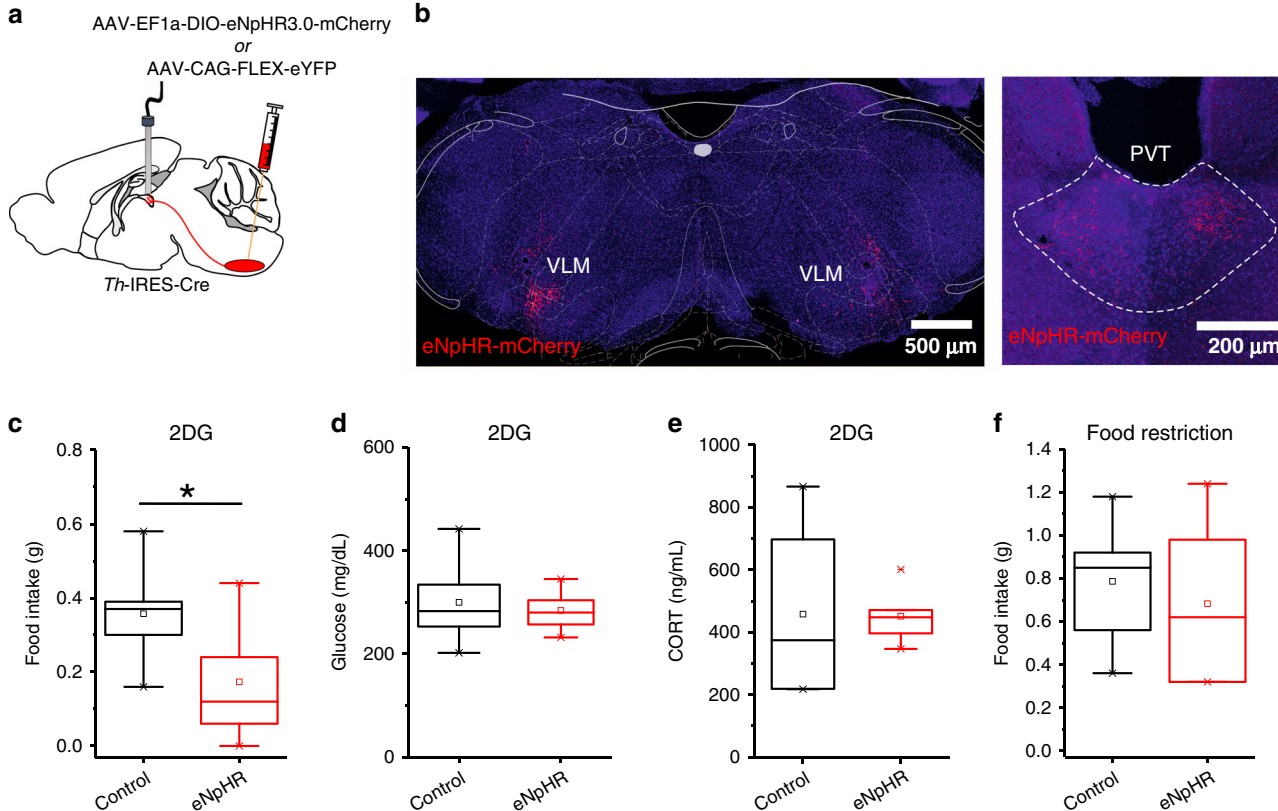

**Fig. 7 The VLM^CA–pPVT pathway is necessary for glucoprivation-induced feeding behavior. a** Schematic of the stereotaxic injections for selective expression of halorhodopsin (eNpHR) and control viral vectors in VLM^CA neurons using of TH-IRES-Cre mice, as well as optical fiber implantation for optogenetic inhibition of VLM^CA projections to the pPVT. **b** Representative image of eNpHR-mCherry expression in the VLM (left) of TH-IRES-Cre mice and eNpHR-mCherry terminals in the pPVT (right). **c** Quantification of food intake 2 h after 2DG administration. TH-IRES-Cre mice expressing eNpHR or control virus were injected with 2DG and immediately subjected to photoinhibition of VLM^CA–pPVT terminals for the first hour of the 2 hr test (light on). VLM^CA–pPVT inhibition suppressed 2DG induced increases in feeding behavior. Total food intake in grams, Control, 0.36 ± 0.04, $n = 8$ mice; eNpHR, 0.17 ± 0.05, $n = 9$ mice; two-sided Unpaired t-test, *$P = 0.018$. **d** Quantification of blood glucose levels after 2DG administration in TH-IRES-Cre mice expressing either eNpHR or a control virus in VLM^CA neurons. VLM^CA–pPVT inhibition had no effect on 2DG-mediated adrenal medullary hyperglycemia. Glucose levels in mg/dL, Control, 300 ± 22.78, $n = 10$ mice; eNpHR, 284.29 ± 13.93, $n = 7$ mice; two-sided Unpaired t-test, $P = 0.61$. **e** Quantification of blood CORT levels after 2DG administration in TH-IRES-Cre mice expressing either eNpHR or a control virus in VLM^CA neurons. VLM^CA–pPVT inhibition had no effect on 2DG increases in corticosterone secretion. CORT levels in mg/dL, Control, 457.90 ± 154.55, $n = 4$ mice; eNpHR, 451.10 ± 29.62, $n = 7$ mice; two-sided Unpaired t-test, $P = 0.96$. **f** Quantification of food intake for 2 h immediately after overnight food restriction. TH-IRES-Cre mice expressing either eNpHR or a control virus were food restricted overnight and then immediately given access to food while subjected to photoinhibition of VLM^CA–pPVT synapses. VLM^CA–pPVT inhibition had no effect on feeding behavior after overnight food restriction, in the absence of 2DG. Total food intake in grams, Control, 0.79 ± 0.12, $n = 6$ mice; eNpHR, 0.68 ± 0.15, $n = 6$ mice; two-sided Unpaired t-test, $P = 0.60$. Box chart legend: box is defined by 25th, 75th percentiles, whiskers are determined by 5th and 95th percentiles, and mean is depicted by the square symbol. Data presented as mean ± SEM.

Vector Biolabs. AAV9-EF1a-FLEX-TVA-mCherry (Addgene plasmid # 38044) and AAV9-CAG-FLEX-RG (Addgene plasmid # 38043) were produced by Vigene Biosciences, Inc. EnvA-SAD-ΔG-eGFP (Addgene plasmid # 32635) was produced by the Viral Vector Core of the Salk Institute for Biological Studies. AAV2-retro-CAG-tdTomato-WPRE is a gift from Dr. Yuanyuan Liu at NIDCR. All viral vectors were stored in aliquots at −80°C until use.

**Drugs**. 2-Deoxy-D-glucose (Tocris catalog# 4515) was used at a 500 mg/Kg dose. CNO (Enzo Life Sciences, catalog# BML-NS105) was used at a 10 mg/Kg dose when used in combination with the inhibitory DREADDs virus and a dose of 0.75 mg/Kg in combination with the excitatory DREADDs virus. Propranolol hydrochloride (Tocris catalog# 0624) at a concentration of 30 μM was used to block beta adrenergic receptors in acute brain slices.

**Antibodies**. The primary antibodies: anti-cFos (1:1000, rabbit polyclonal, Millipore, catalog# ABE457; 1:50, rabbit monoclonal, Cell Signaling, catalog# 2250), validated for application (ABE457 and 2250) and specie (2250); and anti-TH (1:1000, chicken polyclonal, Aves Labs, catalog# AB_10013440), validated for application and specie. Fluorophore-conjugated secondary antibodies (conjugated with goat anti-rabbit Alexa Fluor-488 (A-11008) and goat anti-chicken Alexa Fluor-647 (A-21449)) were purchased from ThermoFisher Scientific and used at a 1:500 dilution. Antibodies were diluted in PBS with 10% NGS and PBST. For

additional information on the validation of the antibodies used, please visit the manufacturer's website.

**Stereotaxic surgery**. All stereotaxic surgeries were conducted as described in our animal study protocol. Mice were first anesthetized with a Ketamine/Xylazine solution and an AngleTwo stereotaxic device (Leica Biosystems) was used for viral injections at the following stereotaxic coordinates: pPVT, −1.60 mm from Bregma, 0.06 mm lateral from midline, and −3.30 mm vertical from cortical surface (at a 6° angle) NAc, 1.70 mm from Bregma, 0.60 mm lateral from midline, and −4.54 mm vertical from cortical surface; VLM, −7.30 mm from Bregma, 1.40 mm from midline, and −5.30 mm vertical from cortical surface. AAVs were injected at a total volume of 0.1–0.15 μl in the VLM. All other AAVs were injected at approximately 1–1.5 μl. Following stereotaxic injections, AAVs were allowed 2–3 weeks for maximal expression. Optical fibers with diameters of 200 μm (0.48 NA) and 400 μm (0.66 NA) were used for optogenetics and fiber photometry experiments, respectively (Doric Lenses). These fibers were implanted over the pPVT immediately after viral injections (coordinate: −1.60 mm from Bregma, 0.06 mm lateral from midline, and −2.90 mm vertical from cortical surface) and cemented using C&B Metabond Quick Adhesive Cement System (Parkell, Inc.) and Jet Brand dental acrylic (Lang Dental Manufacturing Co., Inc.). Mice received subcutaneous injections with metacam (meloxicam, 1–2 mg/kg) for analgesia and anti-

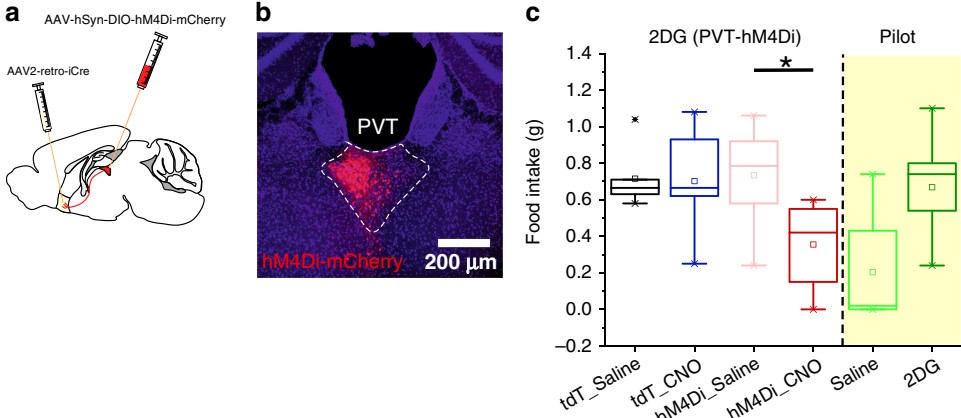

**Fig. 8 pPVT–NAc neuron activity is required for glucoprivation-induced feeding behavior. a** Schematic of the stereotaxic injections for selective expression of the hM4Di-mCherry and control viral vectors in pPVT–NAc neurons. **b** Representative image of hM4Di-mCherry expression in pPVT–NAc neurons. **c** Quantification of food intake for a 5-h period after 2DG administration. Mice expressing either hM4Di or the control virus in pPVT–NAc neurons were injected with either saline or CNO, 30 min prior to 2DG administration. pPVT–Nac neurons inhibition suppressed the 2DG-induced increases in feeding behavior. Total food intake in grams, tdT_Saline, 0.72 ± 0.07, $n = 6$ mice; tdT_CNO, 0.70 ± 0.08, $n = 6$ mice; hM4Di_Saline, 0.73 ± 0.07, $n = 8$ mice; hM4Di_CNO, 0.36 ± 0.09, $n = 8$ mice; effect of treatments, $F_{(1,24)} = 4.35$, $P < 0.05$; effect of virus, $F_{(1,24)} = 3.05$, $P > 0.05$; interaction, $F_{(1,24)} = 3.78$, $P > 0.05$, two-way analysis of variance (ANOVA) followed by Tukey's test. Group comparisons: tdT_Saline vs tdT_CNO, $P = 0.99$, tdT_Saline vs hM4Di_Saline, $P = 1.00$, tdT_Saline vs hM4Di_CNO, $P = 0.06$, tdT_CNO vs hM4Di_CNO, $P = 0.07$; hM4Di_Saline vs hM4Di_CNO, *$P = 0.02$. Data from Supplementary Fig. 8 included for comparison purposes (shaded portion of the graph). Box chart legend: box is defined by 25th, 75th percentiles, whiskers are determined by 5th and 95th percentiles, and mean is depicted by the square symbol. Data presented as mean ± SEM.

inflammatory purposes postoperatively and were allowed to recover on a heating pad where they were constantly monitored.

**Histology**. Mice were injected with euthanasia solution and subsequently sacrificed via transcardial perfusion, first with PBS and then with paraformaldehyde (PFA; 4% in PBS). Brains were then postfixed in 4% PFA at 4 °C overnight, and cryo-protected using a 30% PBS-buffered sucrose solution for ~24–36 h. Coronal brain sections (45 μm) were acquired using a freezing microtome (SM 2010R, Leica). For immunostainings, brain sections were blocked in 10% normal goat serum (NGS) in PBST (0.1% Triton X-100 in PBS) for 1 h at RT, followed by incubation with primary antibodies in 10% NGS-PBST for 48 h at 4 °C. Sections were then washed with PBST (3 × 15 min) and incubated with fluorescent secondary antibodies at RT for 1 h in 10% NGS-PBST. Sections were washed in PBS (3 × 15 min), mounted onto glass slides and coverslipped with ProLong Diamond antifade (ThermoFisher Scientific). Images were taken using a Carl Zeiss LSM 780 confocal microscope using ZEN (version 2.3, Carl Zeiss Microscopy, LLC). Image analysis and cell counting were performed using ImageJ software by a blinded experimenter (Fiji, version 2017 May 30). Optical fiber placements for all subjects included in this study are presented in Supplementary Fig. 11.

**Monosynaptic tracing with pseudotyped rabies virus**. To trace monosynaptic inputs onto NAc-projecting neurons of the PVT we used the pseudotyped rabies virus[20]. Briefly, to limit monosynaptic rabies tracing to NAc-projecting neurons of the PVT, we first injected ~1 μl of the AAV2(retro)-CAG-iCre bilaterally into the NAc of C57BL/6NJ mice. Next, during the same surgical procedure, a virus mixture containing AAV9-EF1a-FLEX-TVA-mCherry and AAV9-CAG-FLEX-RG at a 1:1 ratio was injected into the pPVT (total volume ~1.5 ul), to enable expression of the following components in a Cre-dependent manner: a red fluorescent reporter, mCherry; an avian receptor, termed TVA; and the rabies envelope glycoprotein (G). Two weeks later mice were injected in the same pPVT location with the pseudotyped rabies virus EnvA-SAD-ΔG-eGFP (1.5 μl). Pseudotyping consisted in replacing the envelope glycoprotein G of the rabies virus with an avian envelope termed EnvA. This G-deficient rabies virus can only infect cells that express the TVA receptor, thereby conferring genetic specificity to the method. Importantly, complementation of the pseudotyped rabies virus with envelope glycoprotein in the TVA-expressing cells as described above allows the generation of infectious particles, which in turn can trans-synaptically spread to presynaptic neurons. One week after pseduotyped rabies injection, brain tissues were collected and subjected to analysis.

**Electrophysiology**. For electrophysiological experiments, mice were anesthetized with isoflurane, decapitated, and their brains quickly removed and chilled in ice-cold dissection buffer (110.0 mM choline chloride, 25.0 mM NaHCO₃, 1.25 mM NaH₂PO₄, 2.5 mM KCl, 0.5 mM CaCl₂, 7.0 mM MgCl₂, 25.0 mM glucose, 11.6 mM ascorbic acid, and 3.1 mM pyruvic acid, gassed with 95% O₂ and 5% CO₂). Coronal sections (300 μm thick) containing the pPVT were cut in dissection buffer using a

VT1200S automated vibrating-blade microtome (Leica Biosystems), and were subsequently transferred to incubation chamber containing artificial cerebrospinal fluid (ACSF) (118 mM NaCl, 2.5 mM KCl, 26.2 mM NaHCO₃, 1 mM NaH₂PO₄, 20 mM glucose, 2 mM MgCl₂, and 2 mM CaCl₂, at 34 °C, pH 7.4, gassed with 95% O₂ and 5% CO₂). After at least 40 min recovery time, slices were transferred to room temperature (20–24 °C) and were constantly perfused with ACSF.

Whole-cell patch-clamp recordings from pPVT neurons were obtained with Multiclamp 700B amplifiers (Molecular Devices), under visual guidance with an Olympus BX51 microscope with transmitted light illumination. Current clamp recordings were made with an internal solution containing: 130 mM K-Gluconate, 5 mM KCl, 10 mM HEPES, 2.5 mM MgCl₂, 4 mM Na₂AP, 0.4 mM Na₃GTP, 10 mM Na-phosphocreatine, and 0.6 mM EGTA (pH 7.2). Recorded signals were low-pass filtered at 1 kHz and acquired at a sampling rate of 10 kHz using pCLAMP 10 (Molecular Devices). For patch-clamp recordings of VLM^CA-mediated responses in pPVT neurons, first ChR2 was expressed in the VLM of TH-Cre mice following the above described methods. Optogenetic stimulation was done using an LED illuminator (pE-300 white; CoolLED, Ltd.) directly connected to the epifluorescence port of the Olympus BX51 microscope. Blue light pulses 1 ms long where delivered at 20 Hz for a duration of 5 s. Data from patch-clamp whole-cell recordings were analyzed using Calmpfit (pCLAMP10 suite, Molecular Devices).

**Bulk Ca²⁺ and fiber photometry**. The fiber photometry procedure for calcium measurements were conducted as follows[51]. Briefly, we used a custom-ordered fiber photometry Mini Cube (Doric Lenses) equipped with three excitation and two emission ports. Specifically, the system is integrated with two continuous sinu-soidally modulated LED (DC4100, ThorLabs) at 473 nm (211 Hz) and 405 nm (531 Hz) that served as light source to excite GCaMP6s and an isosbestic auto-fluorescence signal, respectively. Optogenetic manipulations were allowed via a third excitation port (555–570 nm). Fluorescence signals were collected by the same fiber implant that was coupled to a 400 μm optical patch-cord (0.48 NA) and focused onto two separate photoreceivers (2151, Newport Corporation) connected to the emission ports of the Mini Cube. An RZ5P acquisition system (Tucker-Davis Technologies; TDT), equipped with a real-time signal processor and controlled by a software interface (Synapse version 92) was used to control the LEDs and also to independently demodulate the fluorescence signal due to 473 nm and 405 nm excitation. The LED intensity (ranging 10–15 μW) at the interface between the fiber tip and the animal was constant throughout the session. All photometry experiments were performed in behavioral chambers (Coulbourn Instruments) and video recorded using video cameras installed above each behavioral chamber. TTL pulses recorded by the same system were used to annotate the occurrence of behavioral manipulations. For ΔF/F analysis, a least-squares linear fit to the 405 nm signal to align it to the 470 nm signal was first applied. The resulting fitted 405 nm signal was then used to normalize the 473 nm as follows: ΔF/F = (473 nm signal − fitted 405 nm signal)/fitted 405 nm signal. Lastly, changes in fluorescence after stimulation or after feeding were determined by the area under the ΔF/F curve, before and after stimulus.

## Optogenetics and behavioral protocols

*Feeding behavior.* TH-IRES-Cre mice injected with either Cre-dependent ChR2 or Cre-dependent tdTomato (control) in the VLM and an optical fiber placed above the pPVT were behaviorally tested 3 weeks later. First, mice were tethered with an optical patch cord and placed in an open-field box ($45 \times 45 \times 40$ cm) where they were given access to 20 mg food pellets for 30 min (Pre-test). Immediately after the pre-test, mice received light stimulation with a blue laser tuned at 473 nm at a frequency of 20 Hz (10 ms width) for 30 min using a 1 min "ON" 2 min "OFF" protocol. The stimulation protocol used was based on preliminary studies showing constant stimulation produced significant decreases in locomotion (described in the main text). After light stimulation, mice were given another 30 min with access to food (Post-test). For Th-IRES-Cre mice that expressed ChrimsonR virus in lieu of ChR2, a yellow laser tuned at 561 nm, but the stimulation protocol was identical to the one used for ChR2 expressing mice. In addition, for these mice, the duration, quantity, and timing of feeding epochs were quantified using a custom designed feeding experimentation device (FED3) (provided by Alexxai Kravitz). For assessing 2DG-evoked feeding and using halorhodopsin (eNpHR), light stimulation was achieved using a yellow laser (561 nm) delivered in 5 s 'light ON' / 5 s 'light OFF' intervals for 1 hr. This photoinhibition protocol was used to avoid off-target influences of prolonged optogenetic activation (i.e., rebound excitation). The power of both the blue and yellow lasers for all experiments was 5–10 mW, measured at tip of the patch chord.

*Foraging decision task.* The foraging track consisted of an elevated long linear chamber ($150 \times 32 \times 25$, L $\times$ W $\times$ H in cm) that was divided into 3 zones: trigger zone, corridor, and reward zone. The ends of the track were designated as the trigger zone and reward zone, while the center served as the corridor. This task consisted of 60 min sessions of self-paced trials. For each trial, food restricted (85% FFW) mice were trained to wait in the trigger zone for an auditory cue that signaled reward availability. Mice were then required to run from the trigger zone to the reward zone to retrieve a food reward (strawberry Ensure). Lastly, mice returned to the trigger zone to initiate another trial. The movement of the mice was tracked with an overhead camera and by digital distance sensors 15 cm (Pololu Robotics and Electronics) located throughout the track and at the entrance of the reward, trigger zone, and food port. Video tracking was performed using Ethovision XT 12 (Noldus) software. Strawberry Ensure solution was delivered through a liquid dispenser (Med-associates Inc.) localized at the end wall of reward zone, 2 cm above the floor of track. Experimental schedule and data acquisition were implemented through the Abet II software for operant control (Lafayette Instruments Neuroscience) and through the Whisker multimedia software (Lafayette Instruments Neuroscience).

*Real-time place aversion (RTPA).* TH-IRES-Cre mice expressing ChR2 or control virus in the VLM and implanted with an optical fiber in the pPVT were first handled for at least 3 consecutive days. Mice were connected to the fiber patch-cord, then tested for place aversion using a real-time place aversion paradigm. First, mice were allowed to explore a shuttle box containing two adjacent (50 cm×30 cm) chambers for 10 mins to establish a side preference. Thereafter, one side was chosen randomly for optical stimulation, such that constant blue light at a frequency of 20 Hz (10 ms width) and a power of 5–10 mW was delivered upon entry of the preferred chamber. The cumulative time spent in different areas of the shuttle box during baseline and stimulation was tracked, recorded, and quantified using ANY-maze (version 5) behavioral tracking software.

*Elevated zero maze (EZM).* The EZM testing apparatus was made of light-gray plastic and consists of a round path area with 200 cm in circumference. The maze divided into four 50 cm areas (two closed areas and two open areas) and was elevated 50 cm above the floor. The path was 4 cm wide and open areas had a 0.5 cm lip, while close areas had a 10 cm wall enclosure on each section comprised of four 50 cm sections (two opened and two closed). TH-IRES-Cre mice expressing ChR2 in VLM and fiber placed in pPVT and their respective controls were connected to the optical patch-cord and positioned in the inter-section of an open and closed arm. Mice were allowed to explore the maze freely for 10 min. Animals received 20 Hz (10 ms width and 5–10 mW power) photostimulation 1 min "ON", one minute "OFF" stimulation protocol. The cumulative time spent in the closed and open arms of the maze during stimulation was tracked, recorded, and quantified using ANY-maze behavioral tracking software.

*Open field test (OFT).* OFT testing was performed in a square enclosure ($50 \times 50$ cm). We connected TH-IRES-Cre mice to fiber optics and allowed them to roam freely for 10 min. Mice received photostimulation at a frequency of 20 Hz (10 ms width and 5–10 mW power). The cumulative time spent in different areas of the open field apparatus (corners vs center) during stimulation was tracked, recorded, and quantified using ANY-maze behavioral tracking software.

## Blood glucose levels and CORT detection.

Prior to the initiation of the experiments, mice were habituated and handled daily for 5 days. The day after the last habituation session, mice were injected with either saline or 2DG and immediately subjected to optogenetic silencing of the VLM$^{CA}$–pPVT pathway using eNpHR and a photoinhibition protocol identical as those used for the 2DG experiments (light delivered using 5 s 'light ON' / 5 s 'light OFF' intervals for 1 hr; 5–10 mW power). One hour after 2DG injection and optogenetic silencing, blood samples were collected via a small incision in the tip of the subject's tail and blood glucose levels were measured using a Care Touch Diabetes Testing Kit. Immediately after, additional blood samples were collected into a heparinized SafeCrit plastic tube (Tris Sample Processing). Serum was gathered after and centrifuged using a CritSpin® Micro-Hematocrit Centrifuge for 2 min. Corticosterone protein levels from serum were determined using a DetectX® CORTICOSTERONE Enzyme Immunoassay Kit (Arbor Assays) by following manufacturer's protocol. Briefly, serum was dissociated with dissociation reagent for 10 min at room temperature, and then diluted in buffer 1:100. Standards and samples were added into 96-well microplates and followed by adding DetectX® corticosterone conjugate as well as DetectX® conticosterone antibody. After 2 h incubation, wells were washed four times with 300 µl washer buffer. A TMB substrate solution was added into the wells for an additional 30-min incubation without shaking and yielded a blue color, which turned to yellowish when the Stop Solution was added. The optical density of each well was detected using a microplate reader set at 450 nm, and the plate reader's built-in 4PLC software calculated corticosterone concentration for each well.

## Statistics and data presentation.

All data were analyzed using Origin Pro 2016 (OriginLab Corp). All statistical tests are indicated when used. No assumptions or corrections were made prior to data analysis. All data are presented as mean ± s.e.m. All cell counting experiments and behavioral testing were performed by an experimenter blind to the experimental condition. The sample sizes used in our study, such as the numbers of neurons or animals, are typically the same or exceed those estimated by power analysis (power = 0.80, α = 0.05). For electro-physiological analyses, the sample size is 8–15 cells. For optogenetic-only feeding experiments, the sample size is 6–12 mice. For combined optogenetic and fiber photometry experiments, the sample size is 5–9 mice. For fiber photometry experiment in the reward foraging task, the sample size is 5 mice. For optogenetic experiments used to assess aversive and anxiety-like behavior, the sample size is 5–12 mice. For glucose and CORT assays, the sample size was 4–10 mice. For immunohistochemical cFos analyses, the sample size is 3–6 mice. All experiments were replicated at least once.

**Reporting summary.** Further information on research design is available in the Nature Research Reporting Summary linked to this article.

## Data availability

All the data that support the findings presented in this study are available from the corresponding author upon reasonable request. Source data are provided with this paper.

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

## Acknowledgements

We thank C. Beas for his assistance in the design of a script that facilitated the data analysis of the calcium imaging experiments, and the NIMH IRP Rodent Behavioral Core for their help with the affective behavioral tests. This work was supported by the NIMH Intramural Research Program (1ZIAMH002950, to M.A.P.), NIGMS PRAT Fellowship (Fi2GM128811, to B.S.B.) and NIDCR Intramural Research Program (M.H.).

## Author contributions

B.S.B. and X.G. performed anatomical and immunohistochemical studies, calcium experiments, optogenetic experiments, and stereotaxic injections for all experiments. B.S.B, S.R., and O.K. performed all the behavioral experiments. B.S.B. and M.A.P. performed electrophysiological experiments. B.S.B. and Y.L. performed all the blood collection for glucose and CORT measurements. B.S.B., O.K., S.R., and M.K performed histological procedures. M.K. performed cFos quantification. R.S.L. produced the Flp-dependent GCaMP6s vector. B.A.M. and A.K. provided critical reagents and suggestions. B.S.B., X.G., M.H., and M.A.P. designed the study, interpreted results, and wrote the paper.

## Competing interests

The authors declare no competing interests.
