## [Peer Review File · Nature Communications]

Reviewers' Comments:

Reviewer #1:

Remarks to the Author:

This study describes a neuronal circuit formed by catecholaminergic neurons of the ventrolateral medulla (VLM) that project to neurons in the posterior part of the paraventricular thalamus (pPVT), which are themselves connected to the Nucleus Accumbens. Activation of this VLM-pPVT-NAC circuit increases feeding and is responsible for part of the stimulation food intake triggered by 2DG-induced neuroglucopenia. These data characterize, using state-of-the art techniques, the link that has previously been established between neuroglucopenia activation of VLM TH+ neurons and increased feeding. They also show that this pathway does not induce secretion of counterregulatory hormones, indicating that different pathways may link VLM-TH neuron activation to this hormonal response.

This is a very well performed study that presents very convincing data, which are relevant both to the general understanding of the role of the PVT in linking enteroceptive signals, detected at the level of the brainstem, to the control of feeding behavior and in the general brain response to neuroglucopenia.

I have only a few comments.

1. Results, lines 138-153: The observation that stimulation of PVT neurons induces feeding is in contrast to what has been described in others studies (refs 19-22) but more in line with other studies (Refs 23-25). The authors suggest that this may be explained by the fact that "increased PVT activity may be necessary to promote goal-directed behavior like food seeking... but that food consumption itself is instead associated with reduced PVT activity". Although this could be an explanation, it is also possible that different classes of PVT neurons have different impact on feeding as shown, for instance, in Labouébe et al., (Nat. Neurosci. 2016; 19:999) where activation of subpopulation of PVT neurons is shown to have a strong effect on promoting sucrose seeking behavior. Neuronal heterogeneity in the PVT neurons should, thus, also be considered.
2. Figure 2k: it is not clear from this figure whether feeding induces the observed decrease in intracellular Ca⁺⁺ or whether the feeding is induced by the arrest of light activation. It is also possible that the decrease in Ca⁺⁺ may be due to an hyperpolarization of the neurons following the period of light activation. Performing the same experiments in the absence of food would help answering these questions.
3. Line 151-153: the assumption that pPVT neurons drive negative reinforcement is not supported by experimental evidence.

Reviewer #2:

Remarks to the Author:

Beas and colleagues use circuits tracing experiments and behavioral assays to study a medulla-thalamo-striatal circuit for glucose deprivation mediated feeding behavior. They show that activity in neurons of the VLM increases feeding. They also show that activation of VLM can increase activity in PVT-NAC neurons in an adrenergic mediated manner. They also show that inhibition of VLM and PVT-NAC neurons increases feeding in 2DG treated mice which serves as a model of glucodeprivation. Overall the authors are studying a very interesting and timely topic in a circuit that is of high interest. However, some issues with the paper have decreased my enthusiasm including a lack of citation of research that is absolutely critical considering the topic of study (Labouébe et al., 2016), and the lack of clarity and experimental design for establishing circuit connectivity in the VLM-PVT-NAC pathway.

Concerns

1. An issue that I have with this paper is the lack of citation and discussion of critical research related

to the project. For example, Labouébe and colleagues (Labouébe et al 2016 Nat Neurosci 19: 999-1002) showed that PVT-VGLUT2 neurons that project to the ventral striatum become hyperexcitable during hypoglycemia, and optogenetic stimulation of these neurons increases feeding behavior. This is definitely the most important paper to cite in the entire manuscript and should be compared and contrasted extensively such that we can get a better understanding of what the current manuscript adds to the field.

2. In Figure 2, the authors have no data to confirm that VLM-CA neurons directly synapse onto PVT-NAC neurons. For example, the recording solution in Fig 2h-j would have allowed for multisynaptic signals. Voltage-clamp recordings should be performed in the presence of TTX and 4-AP. To be considered for publication, this Figure should also be extensively modified so that it is clearer to the readers of what exactly the authors did in each panel (e.g., see minor concerns below).

Other issues

3. The sentence on line 80: "While chemogenetic activation is a useful tool to investigate the role of defined neurons, it lacks temporal resolution and cannot easily be used to map the synaptic contribution of specific neural outputs" is not necessarily accurate regarding specific outputs. Optogenetics and chemogenetics both have their advantages for studying projection pathways. The following sentence should reiterate that optogenetics improves spatial resolution, rather than neural circuit resolution.

4. Authors combine TH-cre and CAV-cre for labeling of catecholamine and NAC projecting neurons, respectively. Have they confirmed that CAV-cre does not induce expression of non-catecholamine neurons in VLM, or that TH-cre does not induce expression of non-NAC projecting neurons in PVT? Use of a flipase would or another recombinant would be preferable.

5. Data and methods in Figure 2 are poorly described and difficult to understand. For example, in Figure 2h-j, I'm not sure the optogenetic parameters are as they are left out. In Fig 2j, the measurement of change in voltage is unclear. Is this a depolarization? It would be helpful if they plotted as change from RMP rather than %change. I also recommend labeling the panels so that it's easier for the readers to understand what experiment it is, as there are several in this figure.

6. Related to VLM catecholamine neurons, it is unclear of exactly what neurons the authors are targeting. Figure 1e does a poor job of showing the neurons that are labeled following injection of DIO viruses into VLM of TH-cre mice. I suggest quantification of expression within the subnuclei in the medulla so that readers can get a clearer picture of the targeted structure. Alternatively, or in addition, the authors should provide better histological images (zoomed in) showing location of targeted cell bodies for VLM.

7. On a related note, the images in Figure 1a-c, at least to my eyes, do not show Syp-GFP in pPVT as suggested by the authors. This is an important point, and thus either better images or quantification is needed here.

8. Quantified data from Figure S3C would be useful for comparisons sake to be presented in Figure 3. Similarly, data showing feeding in Figure 3G, 3H, 3M in 2DG vs non-2DG treated animals would be useful for comparison.

9. Authors describe studying and targeting VLM catecholamine neurons, but leave no description of what catecholamines these neurons might express except a hint that they might be adrenergic on line 136. Use of TH-cre for targeting CA neurons should be cited and also better described for those who are new to the field.

10. Authors need to clarify and expand discussion of other relevant input and output pathways that

could modulate glucodeprivation-related feeding so that we can get a better understanding of the circuit overall. This includes zona incerta (Zhang 2017), orexin/hypothalamic (Meffre 2019; Choi 2012; many others), PFC (Otis 2017/2019) and LC dopamine (Beas 2018) inputs that could modulate the activity of PVT neurons that project not only to NAC, but also elsewhere in the extended amygdala. This is critical as the chemogenetics findings in Fig 3k-m could be related to PVT-NAC inhibition OR to inhibition of other projection pathways that bifurcate from these neurons.

11. Typos: Line 162.

Reviewer #3:

Remarks to the Author:

In this study, Beas, Gu and colleagues investigate the implication of the VLM projection to the PVT in the control of feeding using a wide diversity of techniques. After showing that the VLM projects to the PVT, they report that optogenetic activation of VLM terminals in PVT induces feeding but not aversion. Using photometry combined to optogenetic, they find that activation of the VLM activate PVT neurons but that feeding was induced after the termination of the stimulation trains, when PVT activity has decreased below baseline levels. They also provide monosynaptic tracing data showing that NAc-projecting PVT neurons receive VLM inputs. A small in-vitro study shows that PVT neurons are mildly depolarized by VLM optogenetic stimulation. In a second set of experiments, this paper describes the effect of VLM chemogenetic and optogenetic inhibition on 2DG-induced feeding.

The topic and the approach are both interesting and timely and the data presented provide a clear advancement in our understanding of the circuitry controlling feeding, especially that coming from the hindbrain. Nonetheless, the paper would greatly benefit from additional analyses of the data and a more careful interpretation.

Major concerns

1. The conclusion of the authors is that VLM stimulation induces feeding. But the data shows the opposite. It is the stimulation termination that induces feeding. The authors justify the use of optogenetics by its superior temporal resolution. Yet, the behavior (feeding) is only presented as aggregated data of ON and OFF stimulation periods. Showing an increase in feeding. Figure S4 does provide more detailed data but lacks the baseline period to compare. A more detailed analysis must be carried out where feeder entries are plotted over time for all experiments. This peculiar effect should be interpreted more deeply in the discussion section that wrongly states that "activation of the VLM [...] promotes robust feeding".

2. In the result section (lines 143-146), the authors propose that PVT stimulation differently affects food-seeking and food consumption, a claim that certainly cannot be made in their case. It is very unlikely that mice take 60s (the stimulation duration) to engage in food-seeking. In order to precise the behavior, could the authors provide some video analysis?

Minor concerns

1. In figure 1, panel d, the virus constructs should be placed over the syringe and not the fiber optic.

2. I found some missing information about the optogenetic experiments.

a. The authors need to provide the light intensity used for all experiments.

b. The method section does not describe the stimulation parameter used for the halorhodopsin experiment (Fig 3e-j).

c. The authors should justify, or at least clearly mention, that the anxiety assays were done with 5ms and not 10ms-pulses.

3. The in-vitro experiment needs to be better described both in the method and the result sections.
 - a. In this experiment, the authors did not identify NAc-projecting PVT neurons but report the effect of VLM stimulation of unidentified PVT neurons. This should be acknowledged in the results.
 - b. What did the author define as an excitation or an inhibition? First, it would seem more accurate to use depolarization and hyperpolarization. But more importantly, what statistical test did the author use to identify significant responses? For figure 2 panel j, I do not understand why the light stimulation decreases rather than increases the voltage? Could the authors present non-normalized data?
 - c. Could the author quantify light-induced spiking in their population?

4. Because PVT neurons are excited by aversive stimuli, the authors propose that PVT drives feeding through negative reinforcement (lines 151-153). Such conclusion is farfetched and does not take into account the fact that many neurons also respond to rewarding stimuli and that their task is devoid of aversive stimuli.

5. Silencing VLM terminals reduced food intake in 2DG-treated but not in food-restricted mice. The authors conclude that 'this circuit is specifically engaged during glucoprivic events'. This is an interesting claim. Could the author provide data on glucose levels in food-restricted mice?

DEPARTMENT OF HEALTH & HUMAN SERVICES
Intramural Research Program
National Institute of Mental Health
Bethesda Maryland 20814

Mario A. Penzo PhD.
Telephone (Office): 301-451-7296
mario.penzo@nih.gov

Dear Reviewers,

Here we send you a revised manuscript entitled “**A ventrolateral medulla-midline thalamic circuit for hypoglycemic feeding**”, which is a revision of the original manuscript NCOMMS-20-16932.

We sincerely thank you for your interest in our study, as well as for your insightful comments and suggestions. For the revision, we conducted additional experiments and data analyses to address all the concerns raised during review and we modified the manuscript’s text to accommodate all suggestions and requested clarifications. These include additional details in the Methods and an extension of the original Discussion section to more appropriately frame our study and expand on the interpretation of our findings. Together, we feel these changes significantly improve the quality of the manuscript. Below is a point-by-point response to comments (in **bold** typeface).

Sincerely,

Mario A. Penzo, Ph.D.

Unit on the Neurobiology of Affective Memory
National Institute of Mental Health

Mark A. Hoon, PhD

Molecular Genetics Section
National Institute of Dental and Craniofacial Research

Major new experiments added:

- 1) We used a different genetic approach that relies on Flp recombinase to express GCaMP6s in NAc-projecting neurons of the pPVT of TH-Cre mice. This approach allowed us to exclude the possibility that injection of a Cre expressing retrograde vector in the NAc results in Cre contamination in non-TH neurons of the VLM, and thereby stimulation of non-TH terminals innervating the pPVT. Whereas, to our knowledge, functional projections from the VLM to the NAc have not been described in the literature, this approach allowed us to further test our hypothesis. Importantly, and consistent with data presented in our initial submission, stimulation of VLM^{CA} terminals elicited significant increases in GCaMP6s fluorescence (**Supplementary Fig. 3a-d**). Similarly, feeding behavior was associated with decreases in fluorescence (**Supplementary Fig. 3e-h**).
- 2) Although data from our initial submission showed that in well-fed mice feeding during our tests was restricted to the optogenetic stimulation block (VLM^{CA}-PVT terminals), the experimental design used to quantify the precise time of occurrence of feeding events relative to optogenetic stimulation did not include pre- or post-tests. Thus, to

satisfy this request, we included these pre- and post-stimulus periods in our Flp-dependent GCaMP6s experiment. Reassuringly, in agreement with our original report, we observed that feeding behavior was mostly restricted to the stimulation block, and typically occurred immediately upon cessation of light stimulation (**Supplementary Fig. 3i, j**).

- 3) To investigate the dynamic of pPVT–NAc neurons during foraging-like food seeking behavior, we used fiber photometry to measure changes in GCaMP6s fluorescence in these neurons while mice performed a reward foraging decision task. Consistent with our model, these findings demonstrate that pPVT–NAc neurons are engaged during reward seeking (approach) and suppressed during reward consumption (**Supplementary Fig. 5**).

New data analyses and changes in data presentation:

- 1) We now present the VLM-induced depolarization of pPVT neurons as change in voltage from resting membrane potential (**Fig. 2j**)
- 2) The terms ‘Activated’ and ‘Inhibited’ in reference to the effect of VLM input stimulation in pPVT neuron, have been replaced by ‘Depolarized’ and ‘Hyperpolarized’, respectively (**Fig. 2i**).
- 3) For comparison purposes, we now include the data showing the effects of chemogenetic activation of VLM^{CA} neurons (hM3Dq) on cFos expression in the pPVT alongside those showing the effects of 2DG (with or without VLM^{CA} neuronal suppression with hM4Di) on the same parameter (**Fig. 3d**). Notice that pPVT cFos expression following 2DG administration in mice in which the activity of VLM^{CA} neurons was suppressed is comparable to that of animals injected with Saline in lieu of 2DG.
- 4) Data from our initial pilot experiment assessing the effects of 2DG and Saline on feeding behavior is now contrasted with those in which 2DG was administered in subjects expressing viral vectors aimed at inhibiting the activity of PVT–NAc neurons (**Fig. 3m**). 2DG-induced feeding behavior following chemogenetic inhibition of pPVT–NAc neurons appeared indistinguishable from feeding behavior observed in Saline-administered mice.
- 5) We now include a representative movie of the feeding behavior induced by optogenetic stimulation of VLM^{CA}–PVT terminals (**Supplementary Movie 1**).
- 6) We have analyzed data from a ‘No-Food’ session where VLM^{CA} afferents to pPVT–NAc neurons were optogenetically stimulated but no access to food was provided. These analyses revealed that food availability resulted in diminished light-evoked responses in pPVT–NAc neurons. Notably, consistent with our previous observation that feeding events are associated with the suppression of pPVT–NAc neuronal activity, we observed a modest inverse correlation between the number pellets consumed during a session and the average GCaMP signal (**Supplementary Fig. 7**).

Reviewer #1 (Remarks to the Author):

This study describes a neuronal circuit formed by catecholaminergic neurons of the ventrolateral medulla (VLM) that project to neurons in the posterior part of the paraventricular thalamus (pPVT), which are themselves connected to the Nucleus Accumbens. Activation of this VLM-pPVT-NAc circuit increases feeding and is responsible for part of the stimulation food intake triggered by 2DG-induced neuroglucopenia. These data characterize, using state-of-the art techniques, the link that has previously been established between neuroglucopenia activation of VLM TH+ neurons and increased feeding. They also show that this pathway does not induce secretion of counterregulatory hormones, indicating that different pathways may link VLM-TH neuron activation to this hormonal response.

This is a very well performed study that presents very convincing data, which are relevant both to the general understanding of the role of the PVT in linking enteroceptive signals, detected at the level of the brainstem, to the control of feeding behavior and in the general brain response to neuroglucopenia.

We thank the reviewer for considering our study to be ‘very well performed’ and to ‘present very convincing data’ that improves the general understanding of the role of the PVT in guiding feeding behavior in response to glucoprivation.

I have only a few comments.

1. Results, lines 138-153: The observation that stimulation of PVT neurons induces feeding is in contrast to what has been described in others studies (refs 19-22) but more in line with other studies (Refs 23-25). The authors suggest that this may be explained by the fact that “increased PVT activity may be necessary to promote goal-directed behavior like food seeking... but that food consumption itself is instead associated with reduced PVT activity”. Although this could be an explanation, it is also possible that different classes of PVT neurons have different impact on feeding as shown, for instance, in Labouèbe et al., (Nat. Neurosci. 2016; 19:999) where activation of subpopulation of PVT neurons is shown to have a strong effect on promoting sucrose seeking behavior. Neuronal heterogeneity in the PVT neurons should, thus, also be considered.

We thank the reviewer for her/his insightful comment. We agree, another potential interpretation of our observations, is that different subpopulations of pPVT–NAc neurons participate in these processes. Indeed, while our recordings are limited to NAc-projecting neurons of the pPVT, it is plausible that this class of pPVT neurons can be further classified into functionally distinct sub-classes. To address this, we have now expanded the Discussion section of our manuscript to include a comment on the possibility of neuronal diversity within the PVT as an alternative interpretation to our finding that pPVT–NAc neurons are bidirectionally modulated by food seeking and consumption (lines 284-292).

2. *Figure 2k: it is not clear from this figure whether feeding induces the observed decrease in intracellular Ca^{++} or whether the feeding is induced by the arrest of light activation. It is also possible that the decrease in Ca^{++} may be due to an hyperpolarization of the neurons following the period of light activation. Performing the same experiments in the absence of food would help answering these questions.*

We thank the reviewer for this comment. For clarification, there seems to be a misunderstanding of the data we present. The change in GCaMP6s fluorescence depicted in Figure 2l-n are time locked to the occurrence of feeding events they do not occur upon the arrest of light stimulation. In fact, as can be observed in sample recordings (Fig. 2k) and datasets (Supplementary Fig. 3i and 6) feeding events often occurred several seconds after the cessation of light stimulation. Yet, time-locking of the GCaMP signal to feeding events revealed an event-induced attenuation in fluorescence (Fig. 2l-n; Supplementary Fig. 3e-h).

Nevertheless, as per the reviewer's request, we now include imaging data from a 'No-Food' session from the same subjects included in Figure 2k-n. During this 'No-Food' session, animals were subjected to the same optogenetic stimulation protocol of VLM^{CA}-pPVT projections but without access to food. This additional data is now included in Supplementary Figure 7. Notice that while termination of light stimulation results in the recovery of GCaMP6s fluorescence in the pPVT in the 'No-Food' session, further decreases in calcium signal were observed when mice had access to food (Food trials) (Supplementary Fig. 7b, d, e). Interestingly, our new analyses also revealed that light-evoked responses were on average larger in the 'No-Food' session than in the 'Food' session (Supplementary Fig. 7a-d). These findings suggest that feeding might negatively modulate pPVT-NAc neuron responsivity to VLM^{CA} input stimulation.

3. *Line 151-153: the assumption that pPVT neurons drive negative reinforcement is not supported by experimental evidence.*

We thank the reviewer for pointing out that in the original manuscript, the comment that our findings “suggest that the pPVT may drive feeding behavior through negative reinforcement” appeared as a direct conclusion of our study. We realize that this statement may lead to confusion since we did not directly evaluate this hypothesis. However, we believe that our new evidence provided (in answer to point 2 above) now lends further support to this hypothesis. Specifically, the above-described observation that access to food during optogenetic stimulation of VLM^{CA}-pPVT projections was associated with attenuated light-evoked responses when compared to the No-Food session (Supplementary Fig. 7). These findings suggest that feeding works to counteract the effects of light stimulation in pPVT-NAc neurons. Considering that activation of VLM^{CA}-pPVT projections is also associated with anxiogenic phenotypes (Supplementary Fig. 1), collectively our results do support the idea the pPVT drives feeding through negative reinforcement. Nonetheless, we have toned the discussion on this point (lines 258-283).

Reviewer #2 (Remarks to the Author):

Beas and colleagues use circuits tracing experiments and behavioral assays to study a medulla-thalamo-striatal circuit for glucose deprivation mediated feeding behavior. They show that activity in neurons of the VLM increases feeding. They also show that activation of VLM can increase activity in PVT-NAC neurons in an adrenergic mediated manner. They also show that inhibition of VLM and PVT-NAC neurons increases feeding in 2DG treated mice which serves as a model of glucodeprivation. Overall the authors are studying a very interesting and timely topic in a circuit that is of high interest. However, some issues with the paper have decreased my enthusiasm including a lack of citation of research that is absolutely critical considering the topic of study (Labouébe et al., 2016), and the lack of clarity and experimental design for establishing circuit connectivity in the VLM-PVT-NAC pathway.

We thank the reviewer for their positive feed-back; ‘very interesting and timely topic in a circuit that is of high interest’. Below we address each one of the reviewer’s critiques. In doing so, we have significantly revised our manuscript .

Concerns

1. An issue that I have with this paper is the lack of citation and discussion of critical research related to the project. For example, Labouébe and colleagues (Labouébe et al 2016 Nat Neurosci 19: 999-1002) showed that PVT-VGLUT2 neurons that project to the ventral striatum become hyperexcitable during hypoglycemia, and optogenetic stimulation of these neurons increases feeding behavior. This is definitely the most important paper to cite in the entire manuscript and should be compared and contrasted extensively such that we can get a better understanding of what the current manuscript adds to the field.

We thank the reviewer for pointing out an important study that was inadvertently excluded from our list of cited work (Labouébe et al., 2016 Nat Neurosci 19: 999-1002). We have revised our manuscript and now discuss this study both in the Results and Discussion sections.

As described by the reviewer, Labouébe et al. showed that neurons that project to the NAc become activated during hypoglycemic conditions induced *in vitro*. The study goes on to show that these responses depend on the expression of the glucose transporter Glut2 by a subclass of PVT–NAc neurons. Last, the study demonstrates that optogenetic stimulation of these Glut2-expressing neurons *in vivo* can improve performance in a reward task. Collectively, these findings suggest that PVT neurons are equipped with intrinsic mechanisms that could facilitate food seeking behavior in response to glucose deficits.

While this is an interesting study that adds to a growing body of literature implicating the PVT in reward related behaviors^{1,2}, various important questions surrounding this studies’s

main findings. For instance, it is unclear whether activation of GluT2 neurons is sufficient to drive feeding behavior in well fed mice. Similarly, is GluT2-dependent modulation of PVT–NAc neurons required for hypoglycemic feeding? As such, it is difficult to determine the precise contributions of the cellular and circuit mechanism described by this study on feeding behavior associated with glucopenia.

In contrast, here we linked a previously neglected projection from VLM^{CA} neurons to the pPVT that drives robust feeding behavior in response extreme glucose deficits. Our findings demonstrate that activation of the VLM^{CA}–PVT–NAc pathway is not only required for glucoprivic feeding but also sufficient to promote robust feeding in sated mice. As such, as we now comment in the Discussion, we believe that, in addition to the intrinsic glucose sensing mechanisms of PVT neurons, the existence of parallel hypothalamus–PVT and VLM^{CA}–pPVT circuits may constitute fail-safe mechanisms that ensure appropriate adaptation to metabolic status’ (lines 319-329).

2. In Figure 2, the authors have no data to confirm that VLM-CA neurons directly synapse onto PVT-NAc neurons. For example, the recording solution in Fig 2h-j would have allowed for multisynaptic signals. Voltage-clamp recordings should be performed in the presence of TTX and 4-AP. To be considered for publication, this Figure should also be extensively modified so that it is clearer to the readers of what exactly the authors did in each panel (e.g., see minor concerns below).

We agree with the reviewer that in the absence of TTX and 4-AP in the bath solution, voltage clamp recordings alone cannot conclusively demonstrate the existence of monosynaptic connections. However, we present data in Supplementary Figure 2 where we used a monosynaptic rabies tracing approach to assess whether VLM^{CA} neurons form monosynaptic connections onto NAc-projecting neurons of the pPVT. Briefly, we injected a retro-Cre virus in the NAc and subsequently a rabies starter virus expressing the rabies glycoprotein and the avian receptor TVA in the pPVT. After 14 days, we injected pseudotyped rabies virus in the pPVT. The rabies virus was pseudotyped with an avian envelope protein, EnvA, that binds exclusively to TVA. Using this approach, we observed rabies expressing neurons in the VLM that were subsequently identified as TH+ via post-hoc immunostaining with Anti-TH antibody. Based on extensive literature, we believe that the most parsimonious interpretation from this tracing experiment is that VLM^{CA} neurons form monosynaptic connections onto NAc-projecting neurons of the pPVT.

Other issues

3. The sentence on line 80: “While chemogenetic activation is a useful tool to investigate the role of defined neurons, it lacks temporal resolution and cannot easily be used to map the synaptic contribution of specific neural outputs” is not necessarily accurate regarding specific outputs. Optogenetics and chemogenetics both have their advantages for studying projection pathways. The following sentence should reiterate that optogenetics improves spatial resolution, rather than neural circuit resolution.

We thank the reviewer for this recommendation and edited this sentence appropriately (lines 78-82).

4. Authors combine TH-cre and CAV-cre for labeling of catecholamine and NAC projecting neurons, respectively. Have they confirmed that CAV-cre does not induce expression of non-catecholamine neurons in VLM, or that TH-cre does not induce expression of non-NAC projecting neurons in PVT? Use of a flipase would or another recombinant would be preferable.

We thank the reviewer for critically evaluating our data and for offering valuable insight. We would like to clarify that to the best of our knowledge no study has identified functional projections from the VLM to the NAc, or ectopic expression of Cre recombinase in PVT neurons of TH-IRES-Cre knock-in mice. However, we considered the reviewer's suggestion to be a unique opportunity to further test our model and predictions. As such, we now include a new experiment in which retrograde Flp recombinase and Flp-dependent GCaMP6s were used to image the activity of NAc-projecting pPVT neurons while exciting VLM^{CA}-pPVT projections using ChrimsonR expressed in the VLM of TH-Cre mice (Supplementary Fig. 3 and lines 132-142 Main text). This approach affirmed our initial conclusions. Specifically, optogenetic stimulation of VLM^{CA}-PVT afferents led to robust feeding behavior in sated mice, and this was associated with a light-induced increase, and subsequent food consumption related decrease, in GCaMP6s fluorescence in pPVT-NAc neurons (Supplementary Fig. 3)

5. Data and methods in Figure 2 are poorly described and difficult to understand. For example, in Figure 2h-j, I'm not sure the optogenetic parameters are as they are left out. In Fig 2j, the measurement of change in voltage is unclear. Is this a depolarization? It would be helpful if they plotted as change from RMP rather than %change. I also recommend labeling the panels so that it's easier for the readers to understand what experiment it is, as there are several in this figure.

We thank the reviewer for pointing out that the initial description of the data presented in Figure 2 and their associated methods were unclear. We realize that this is a rather large figure with multiple panels. As per the reviewer's suggestion, we added additional clarification by expanding our description of the data presented in this figure in the main text as well as in the figure legend. Furthermore, we also included additional labels for panels to aid the reader.

6. Related to VLM catecholamine neurons, it is unclear of exactly what neurons the authors are targeting. Figure 1e does a poor job of showing the neurons that are labeled following injection of DIO viruses into VLM of TH-cre mice. I suggest quantification of expression within the subnuclei in the medulla so that readers can get a clearer picture of the targeted structure. Alternatively, or in addition, the authors should provide better histological images (zoomed in) showing location of targeted cell bodies for VLM.

Classically, the VLM is thought to contain two groups of catecholaminergic neurons, termed the C1 and A1 cell groups. Our experiments aimed to achieve robust expression of transgenes in both the A1/C1. Whereas C1 neurons (which are adrenergic) can be distinguished from their A1 neighbors (which are noradrenergic) by their expression of the enzyme phenylethanolamine-*N*-methyltransferase (PNMT)³. Indeed, 2-DG induced hypoglycemia has been shown to activate of both A1 and C1 neuronal population. Therefore, we did not distinguish between these two subgroups. We have clarified this issue in the manuscript's Discussion (lines 263-267). In addition, we included a magnified image for better appreciation of the location of these neurons (Fig. 1c).

7. On a related note, the images in Figure 1a-c, at least to my eyes, do not show Syp-GFP in pPVT as suggested by the authors. This is an important point, and thus either better images or quantification is needed here.

We modified the images (possibly images were compressed in the merged version of the manuscript such that details present in the originals were not visible); Syp-GFP expression in the pPVT is depicted by the yellow arrow labeled '1' in Figure 1b (left panel). In addition, Syp-GFP expression can be seen embedded within the Anti-TH labeling and tdTomato-labeled cells in the pPVT in Figure 1c (left panel). In addition, we include an additional arrow in Figure 1c that points at Syp-GFP expression in the magnified image.

8. Quantified data from Figure S3C would be useful for comparisons sake to be presented in Figure 3. Similarly, data showing feeding in Figure 3G, 3H, 3M in 2DG vs non-2DG treated animals would be useful for comparison.

As requested by the reviewer, we now include quantified data from Supplementary Figure 3c in Figure 3. However, we feel that comparing feeding data from Fig. 3g (eNpHR experiment) with our pilot non-2DG (saline) treatment would not be appropriate because of the distinct temporal profiles of each test. While feeding following 2DG treatment is traditionally assessed for a period of 5 hr post injection (see Supplementary Fig. 8), for our optogenetic silencing experiment in Fig. 3g, feeding was quantified only during the first two hours post 2-DG administration. This was intended to better match the temporal profile of the time-dependent optogenetic silencing experiments we performed.

Because CNO is known to elicit effects that can last up to 10 hr⁴, food intake following 2-DG administration in this experiment (Fig. 3m) was compared to pilot data (including saline treatment) presented in Supplementary Fig. 8 (Fig. 3m).

9. Authors describe studying and targeting VLM catecholamine neurons, but leave no description of what catecholamines these neurons might express except a hint that they might be adrenergic on line 136. Use of TH-cre for targeting CA neurons should be cited and also better described for those who are new to the field.

We thank the reviewer for pointing this out and we have tried to clarify this point in our revision. As described above (point 5), the VLM contains two groups of catecholaminergic neurons termed the A1 and C1 cell groups. Whereas C1 neurons are adrenergic and express both dopamine- β -hydroxylase (DBH) and PNMT, A1 cells are noradrenergic and only express DBH. Both DBH-Cre and TH-Cre animals have been recently manipulated to explore the role of these neurons in metabolic regulation. Citation of some of these seminal studies as well as additional rationale on the use of TH-Cre mice is included in our revised manuscript (lines 60-64).

10. Authors need to clarify and expand discussion of other relevant input and output pathways that could modulate glucodeprivation-related feeding so that we can get a better understanding of the circuit overall. This includes zona incerta (Zhang 2017), orexin/hypothalamic (Meffre 2019; Choi 2012; many others), PFC (Otis 2017/2019) and LC dopamine (Beas 2018) inputs that could modulate the activity of PVT neurons that project not only to NAC, but also elsewhere in the extended amygdala. This is critical as the chemogenetics findings in Fig 3k-m could be related to PVT-NAC inhibition OR to inhibition of other projection pathways that bifurcate from these neurons.

We thank the reviewer for this recommendation. We have now expanded our Discussion to comment on the contribution of other afferent inputs to the PVT that may participate in regulating appetitive behaviors (lines 293-310).

11. Typos: Line 162.

We corrected this typo.

Reviewer #3 (Remarks to the Author):

In this study, Beas, Gu and colleagues investigate the implication of the VLM projection to the PVT in the control of feeding using a wide diversity of techniques. After showing that the VLM projects to the PVT, they report that optogenetic activation of VLM terminals in PVT induces feeding but not aversion. Using photometry combined to optogenetic, they find that activation of the VLM activate PVT neurons but that feeding was induced after the termination of the stimulation trains, when PVT activity has decreased below baseline levels. They also provide monosynaptic tracing data showing that NAc-projecting PVT neurons receive VLM inputs. A small in-vitro study shows that PVT neurons are mildly depolarized by VLM optogenetic stimulation. In a second set of experiments, this paper describes the effect of VLM chemogenetic and optogenetic inhibition on 2DG-induced feeding.

The topic and the approach are both interesting and timely and the data presented provide a clear advancement in our understanding of the circuitry controlling feeding, especially that coming from the hindbrain. Nonetheless, the paper would greatly benefit from additional analyses of the data and a more careful interpretation.

We thank the reviewer for noting that data presented ‘provide a clear advancement in our understanding of the circuitry controlling feeding, especially that coming from the hindbrain’. Below we address each of the reviewer’s critiques.

Major concerns

1. The conclusion of the authors is that VLM stimulation induces feeding. But the data shows the opposite. It is the stimulation termination that induces feeding. The authors justify the use of optogenetics by its superior temporal resolution. Yet, the behavior (feeding) is only presented as aggregated data of ON and OFF stimulation periods. Showing an increase in feeding. Figure S4 does provide more detailed data but lacks the baseline period to compare. A more detailed analysis must be carried out where feeder entries are plotted over time for all experiments. This peculiar effect should be interpreted more deeply in the discussion section that wrongly states that “activation of the VLM [...] promotes robust feeding”.

We thank the reviewer for pointing out the peculiarity of the behavioral phenotype associated with stimulation of VLM^{CA}–PVT projections. As the reviewer notes, data shown in Figure 2 and Supplementary Figure 6 (previously Supplementary Fig. 4) demonstrates that feeding events typically occur immediately following, and not during, optogenetic stimulation of this pathway. To illustrate this behavioral response further, we now include a sample video (Supplementary Movie 1). Notice that in the video while mice do not seek food during a pre-test period, food seeking behavior was reliably triggered immediately upon cessation of light stimulation.

With the exception of our fiber photometry imaging experiments, we do not have the precise timing for individual feeding events. Note that gathering this information required implementation of a specialized feeding device (FED3, See Methods) equipped with an analog output that synchronized feeding events to light stimulation and the GCaMP6s signal. However, because we now include a variation of our fiber photometry experiment (in Fig. 2) in which GCaMP6s is expressed in a Flp-dependent manner (to avoid for potential Cre contamination in non-TH VLM neurons), we conducted evaluation of feeding bout timings during both the pre- and post-test periods (Supplementary Fig. 3). As shown in this new figure, feeding events almost exclusively occurred during the light stimulation block. Notably, consistent with our previous data, feeding events largely occurred during the light-off periods of the stimulation blocks.

2. In the result section (lines 143-146), the authors propose that PVT stimulation differently affects food-seeking and food consumption, a claim that certainly cannot be made in their case.

As indicated by the reviewer, we initially speculated that PVT activity may be bidirectionally modulated by food seeking and consumption. However, such interpretation is precluded by the lack of empirical evidence supporting its validity. Thus, to formally test this prediction we designed and implemented a reward foraging decision task that allowed

for investigating the dynamics of PVT–NAC neuronal activity during both reward approach and consummation. Briefly, as described in the manuscript, mice were trained to navigate a linear track in which visiting a “trigger-zone” located on one end of a track enabled access to a reward on the opposite end (“reward-zone”). For each trial, reward availability was signaled via a 1 s 8 KHz tone presented when subjects remained on the “trigger-zone” for a least 2 s. Mice readily learned to perform this task, as exemplified by their ability to complete 40-50 trials per session by the end of training. Importantly, using fiber photometry imaging we discovered that the activity of PVT–NAC neurons increases as mice approach the reward zone and rapidly decreases during reward consumption (Supplementary Figure 5). Collectively, these results demonstrate that, at the population level, the activity of PVT–NAC neurons is bidirectionally modulated during reward approach and consumption.

It is very unlikely that mice take 60s (the stimulation duration) to engage in food-seeking. In order to precise the behavior, could the authors provide some video analysis?

As mentioned above, we now provide a sample video illustrating the behavioral phenotype (Supplementary Movie 1). Consistent with data presented in Supplementary Figure 6, mice do typically wait until the cessation of light stimulation to engage in feeding behavior. Anecdotally, our initial discovery of the feeding phenotype associated with stimulation of the VLM^{CA}–pPVT projections came as a result of a pilot study where we stimulated this projection while mice navigated an open field. Upon removing subjects from the arena, we noticed that they routinely started to consume chow while sitting on the wire top and prior to removal of the patch cord. The reproducibility of this effect inspired us to formally assess the feeding phenotype.

While collectively our studies demonstrate that stimulation of VLM^{CA}–pPVT projections is *sufficient* to produce subsequent robust feeding behavior, and more importantly that activity in this pathway is *required* for glucoprivic feeding, it is unclear whether the precise feeding dynamics associated with optogenetic stimulation of this circuit largely results in non-physiological patterns of activity. An alternative interpretation is that optogenetic stimulation of the VLM^{CA}–pPVT pathway is associated with a negative state such as that produced by hypoglycemia (including dizziness, confusion and anxiety), which may momentarily prevent mice from immediately engaging in feeding behavior due to its debilitating nature. Sudden removal of the robust light-mediated stimulation of this circuit may lessen the severity of the experience such that mice can engage in feeding behavior. While mostly speculative, our observation that stimulation of VLM^{CA}–pPVT projections is associated with decreased locomotion and increased anxiety partially support this interpretation.

In our revised manuscript we speculate on potential interpretations surrounding the dynamics of feeding behavior associated with stimulation of the VLM^{CA}–pPVT pathway in the Discussion.

Minor concerns

1. In figure 1, panel d, the virus constructs should be placed over the syringe and not the fiber optic.

We corrected this mistake.

2. I found some missing information about the optogenetic experiments.
a. The authors need to provide the light intensity used for all experiments.

This information has been added to the Methods.

b. The method section does not describe the stimulation parameter used for the halorhodopsin experiment (Fig 3e-j).

We have added the missing information to the Methods.

c. The authors should justify, or at least clearly mention, that the anxiety assays were done with 5ms and not 10ms-pulses.

The same pulse duration (10 ms) was used for all *in vivo* ChR2 and ChrimsonR experiments. This information has been corrected in the Methods. 1ms light pulses were used for *in vitro* recordings, this is now clarified in Figure 2.

3. The *in-vitro* experiment needs to be better described both in the method and the result sections.
a. In this experiment, the authors did not identify NAc-projecting PVT neurons but report the effect of VLM stimulation of unidentified PVT neurons. This should be acknowledged in the results.

We have now clarified this in the main text.

b. What did the author define as an excitation or an inhibition? First, it would seem more accurate to use depolarization and hyperpolarization. But more importantly, what statistical test did the author use to identify significant responses? For figure 2 panel j, I do not understand why the light stimulation decreases rather than increases the voltage? Could the authors present non-normalized data?

We thank the reviewer for this valuable suggestion. We have replaced the terms ‘Excited’ and ‘Inhibited’ with the more appropriate ‘Depolarized’ and ‘Hyperpolarized’, respectively. In addition, as it is now clarified in the figure legend, and we defined light evoked depolarizations and hyperpolarizations as changes of at least 3 standard deviations

from baseline in the membrane potential. Finally, as per the reviewer's request, we now present the data as voltage change from resting membrane potential (Fig. 2j).

c. Could the author quantify light-induced spiking in their population?

In our system, light-induced spiking largely depended on the resting membrane potential of the recorded neurons. For instance, in slightly hyperpolarized cells (approx. -65mV) light stimulation often induced low-threshold spikes and bursting. In depolarized cells (approx. -50 mV), light stimulation elicited tonic firing activity in many trials. However, for the majority of cells, optogenetic stimulation of VLM^{CA} terminals did not produce reliable spiking behavior. As such, we consider that quantifying light-evoked spiking is not appropriate.

4. Because PVT neurons are excited by aversive stimuli, the authors propose that PVT drives feeding through negative reinforcement (lines 151-153). Such conclusion is farfetched and does not take into account the fact that many neurons also respond to rewarding stimuli and that their task is devoid of aversive stimuli.

We thank the reviewer for pointing out that in the original manuscript, the comment that our findings “suggest that the pPVT may drive feeding behavior through negative reinforcement” appeared as a direct conclusion of our study. We realize that this statement led to confusion since our study did not directly evaluate this hypothesis. However, we believe that new evidence provided in our revised manuscript lends further support to this hypothesis. Please see response to Reviewer 1, point 3.

5. Silencing VLM terminals reduced food intake in 2DG-treated but not in food-restricted mice. The authors conclude that ‘this circuit is specifically engaged during glucoprivic events’. This is an interesting claim. Could the author provide data on glucose levels in food-restricted mice?

Unfortunately, we did not collect glucose levels in food restricted mice. However, it is important to note that previous studies looking at the effect of overnight fasting on blood glucose level describe only moderate changes in blood glucose⁵ (~5.0 mmol/l). In contrast VLM^{CA} neurons are recruited under conditions of hypoglycemia⁶ (~2.5 mmol/l).

References:

- 1 Otis, J. M. *et al.* Paraventricular thalamus projection neurons integrate cortical and hypothalamic signals for cue-reward processing. *Neuron* **102**, doi:<https://doi.org/10.1016/j.neuron.2019.05.018>. (2019).
- 2 Meffre, J. *et al.* Orexin in the Posterior Paraventricular Thalamus Mediates Hunger-Related Signals in the Nucleus Accumbens Core. *Curr Biol*, doi:10.1016/j.cub.2019.07.069 (2019).

DEPARTMENT OF HEALTH & HUMAN SERVICES

Intramural Research Program
National Institute of Mental Health
Bethesda Maryland 20814

Mario A. Penzo PhD.
Telephone (Office): 301-451-7296
mario.penzo@nih.gov

- 3 Ritter, S., Llewellyn-Smith, I. & Dinh, T. T. Subgroups of hindbrain catecholamine neurons are selectively activated by 2-deoxy-D-glucose induced metabolic challenge. *Brain Res* **805**, 41-54, doi:10.1016/s0006-8993(98)00655-6 (1998).
- 4 Farrell, M. S. & Roth, B. L. Pharmacosynthetics: Reimagining the pharmacogenetic approach. *Brain Res* **1511**, 6-20, doi:10.1016/j.brainres.2012.09.043 (2013).
- 5 Jensen, T. L., Kiersgaard, M. K., Sorensen, D. B. & Mikkelsen, L. F. Fasting of mice: a review. *Lab Anim* **47**, 225-240, doi:10.1177/0023677213501659 (2013).
- 6 Jokiahho, A. J., Donovan, C. M. & Watts, A. G. The rate of fall of blood glucose determines the necessity of forebrain-projecting catecholaminergic neurons for male rat sympathoadrenal responses. *Diabetes* **63**, 2854-2865, doi:10.2337/db13-1753 (2014).

Reviewers' Comments:

Reviewer #1:

Remarks to the Author:

This revised version of the manuscript fully addresses my concerns and I don't have any additional request.

Reviewer #2:

Remarks to the Author:

The authors did a good job of addressing my concerns. Well done.

Although the completed experiments have alleviated those concerns, I do think that Reviewer 3 raises some important points that I missed - - and they should be thoroughly addressed before publication. I discuss the specifics below.

It does seem that animals feed after photo stimulation of VLM>>PV neurons, rather than during photo stimulation. In addition, it would be nice to see time course data for photo inhibition of this pathway, such that we can understand the timing of the behavioral effects for that experiment. These points are quite important, as optogenetic "smashing" of the neurons is going to cause the opposite neurophysiological effect upon light cessation due to rebound hyperpolarization (in ChR2 experiments) and spiking (in eNPHR experiments). The authors have not sufficiently addressed these timing concerns brought up by Reviewer 3.

On the other hand, the authors show very convincingly that chemogenetic manipulations of this circuitry cause behavioral effects that are similar to their optogenetic studies, providing strong support for their conclusions.

Another interesting point that the authors make is the possibility of PV-NAC neurons differentially modifying food seeking and consumption. This point, in addition to the above points, highlights the complex nature of PV in regulating motivated behaviors. Specifically, there are diverse behavioral effects found due to activation and inhibition PV cell types, both specific and nonspecific (as recently reviewed by Jackie McGinty's lab in *Frontiers in Behavioral Neuroscience*; link below). Thus, although the authors provide a significant advance in our understanding here, I still believe that a lot is unknown about how VLM influences specific neuronal outputs in PV to influence feeding. Authors should further highlight this complexity such that readers fully understand the diverse nature of PV.

However, for the purposes of this manuscript, I feel that only minor edits are necessary to address Reviewer 3's concerns. This paper will then be suitable for publication in *Nature Communications*.

Link to McGinty Review:

https://www.frontiersin.org/articles/10.3389/fnbeh.2020.590528/full?utm_source=F-NTF&utm_medium=EMLX&utm_campaign=PRD_FEOPS_20170000_ARTICLE&fbclid=IwAR089wQVqJ4b eNKw1sWw35rgXx_dE82_4e76hP9B0rihG0KwnTm-pF6mS8s

DEPARTMENT OF HEALTH & HUMAN SERVICES
Intramural Research Program
National Institute of Mental Health
Bethesda Maryland 20814

Mario A. Penzo PhD.
Telephone (Office): 301-451-7296
mario.penzo@nih.gov

Dear Reviewers,

Here we send you the final version of our manuscript entitled “**A ventrolateral medulla-midline thalamic circuit for hypoglycemic feeding**”, which is a revision of manuscript NCOMMS-20-16932A.

We sincerely thank you for your interest in our study, as well as for your insightful comments and suggestions. Below, please find a point-by-point response to the remaining critiques raised by Reviewer 2 (in **bold** typeface).

Sincerely,

Mario A. Penzo, Ph.D.
Unit on the Neurobiology of Affective Memory
National Institute of Mental Health

Mark A. Hoon, PhD
Molecular Genetics Section
National Institute of Dental and Craniofacial Research

Response to critiques raised by Reviewer 2:

Reviewer #2 (Remarks to the Author):

The authors did a good job of addressing my concerns. Well done.

We are pleased with the Reviewer’s assessment that the initial critiques raised by the reviewer have been addressed in full.

Although the completed experiments have alleviated those concerns, I do think that Reviewer 3 raises some important points that I missed - - and they should be thoroughly addressed before publication. I discuss the specifics below.

It does seem that animals feed after photo stimulation of VLM>>PV neurons, rather than during photo stimulation. In addition, it would be nice to see time course data for photo inhibition of this pathway, such that we can understand the timing of the behavioral effects for that experiment. These points are quite important, as optogenetic "smashing" of the neurons is going to cause the opposite neurophysiological effect upon light cessation due to rebound hyperpolarization (in ChR2 experiments) and spiking (in eNPHR experiments). The authors have not sufficiently addressed these timing concerns brought up by Reviewer 3.

We agree with the reviewer that the behavioral dynamics resulting from the optogenetic stimulation of VLM^{CA}-pPVT terminals are complex. However, while potential hyperpolarization of pPVT-NAc neurons following light stimulation could impact the interpretation of our data, we would like to point out that this concern has already been addressed by data included in the previous submission.

Specifically, Supplementary Fig. 7 shows that cessation of optogenetic stimulation of VLM^{CA} input to the pPVT leads to a return to baseline in GCaMP6s fluorescence in pPVT-NAc neurons (in No-Food trials). This indicates that, at the population level, Light-Off dynamics in the GCaMP6s fluorescence of PVT-NAc neurons are not consistent with afterhyperpolarization due to optogenetic smashing. Indeed, below baseline decreases in GCaMP6s fluorescence after Light-Off were only apparent in Food trials, consistent with our recurrent observation that unlike food seeking behavior (Supplementary Fig. 5a-e), feeding itself results in the suppression of pPVT-NAc neuron activity (Fig. 4 and 5, Supplementary Fig. 3f-h and 7).

On the other hand, the authors show very convincingly that chemogenetic manipulations of this circuitry cause behavioral effects that are similar to their optogenetic studies, providing strong support for their conclusions.

We thank the reviewer for highlighting this very important point. Altogether, our results demonstrate that 1) stimulation of VLM^{CA} input to the PVT is *sufficient* to promote feeding behavior in sated mice, and 2) activation of the VLM^{CA}-pPVT-NAc pathway is *required* for glucoprivic feeding. Considering that experiments involving optogenetic excitation could result in behaviors that lie outside of the animal's natural repertoire (Krakauer et al., 2017), our findings that the pathway described here is selectively recruited to promote glucoprivic feeding answer an important gap in the literature.

*Another interesting point that the authors make is the possibility of PV-NAC neurons differentially modifying food seeking and consumption. This point, in addition to the above points, highlights the complex nature of PV in regulating motivated behaviors. Specifically, there are diverse behavioral effects found due to activation and inhibition PV cell types, both specific and nonspecific (as recently reviewed by Jackie McGinty's lab in *Frontiers in Behavioral Neuroscience*; link below). Thus, although the authors provide a significant advance in our understanding here, I still believe that a lot is unknown about how VLM influences specific neuronal outputs in PV to influence feeding. Authors should further highlight this complexity such that readers fully understand the diverse nature of PV.*

We agree with the reviewer that the neuronal circuits of the PVT are largely heterogeneous, and that this heterogeneity likely underlies the PVT's complex contribution to emotional and motivated behaviors. Indeed, in a recent study from our group we identified two genetically, anatomically and functionally distinct classes of PVT neurons (Gao et al., 2020). As such, we have expanded our manuscript's Discussion to frame our observations within this context and to cite the recent review article by McGinty and Otis

DEPARTMENT OF HEALTH & HUMAN SERVICES

Intramural Research Program
National Institute of Mental Health
Bethesda Maryland 20814

Mario A. Penzo PhD.
Telephone (Office): 301-451-7296
mario.penzo@nih.gov

(lines 283-290).

However, for the purposes of this manuscript, I feel that only minor edits are necessary to address Reviewer 3's concerns. This paper will then be suitable for publication in Nature Communications.

References:

Krakauer JW, Ghazanfar AA, Gomez-Marin A, MacIver MA, Poeppel D. Neuroscience Needs Behavior: Correcting a Reductionist Bias. *Neuron*. 2017 Feb 8;93(3):480-490. doi: 10.1016/j.neuron.2016.12.041. PMID: 28182904.

Gao C, Leng Y, Ma J, Rooke V, Rodriguez-Gonzalez S, Ramakrishnan C, Deisseroth K, Penzo MA. Two genetically, anatomically and functionally distinct cell types segregate across anteroposterior axis of paraventricular thalamus. *Nat Neurosci*. 2020 Feb;23(2):217-228. doi: 10.1038/s41593-019-0572-3. Epub 2020 Jan 13. PMID: 31932767; PMCID: PMC7007348.